# AVI-Bench: Toward Human-like Audio-Visual Intelligence of Omni-MLLMs

Yaoting Wang [1]   Ziyi Zhang [2]   Wenming Tu [3]   Shaoxuan Xu [4]   Wenjie Du [5]   Cheng Liang [3]
Weijun Wang [6]   Yuanchao Li [7]   Guangyao Li [8]   Hao Fei [9]   Yuanchun Li [6]   Henghui Ding [†1]   Yunxin Liu [6]

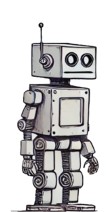 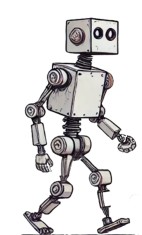 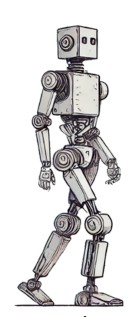 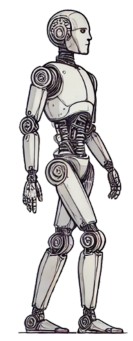

**Task Adaptive**
Models demonstrate effective overall performance across a wide range of audio-visual tasks.

**Modal Adaptive**
Models demonstrate strong performance on both audio and visual modalities.

**Stage Adaptive**
Models illustrate strong performance on both perception and understand for better audio-visual reasoning.

**Domain Adaptive**
Models show human-like domain adaptation.

*Figure 1.* The AVI taxonomy and what it reveals. AVI-Bench arranges audio-visual intelligence into four nested levels: per-task performance (Task Adaptive, Section 5.1), cross-modal balance (Modal Adaptive, Section 5.2), cognitive-stage composition (Stage Adaptive, Section 5.3), and unfamiliar-domain adaptation (Domain Adaptive, Section 5.4). Each level isolates a distinct failure mode hidden by aggregate evluation.

## Abstract

Recent advances in Omni-Multimodal Large Language Models (Omni-MLLMs) have enabled strong integration of vision, audio, and language. However, their audio-visual intelligence (AVI) remains insufficiently evaluated due to the lack of systematic and comprehensive benchmarks. We introduce **AVI-Bench**, a cognitively inspired benchmark that evaluates Omni-MLLMs across three stages, perception, understanding, and reasoning, through cross-modal tasks requiring joint audio-visual interpretation. This design enables fine-grained diagnosis of model capabilities and failure modes. To further assess robustness beyond familiar domains, we propose **AVI-Bench-PriSe**, an extension that probes models' primitive audio-visual sensation using unfamiliar, low-semantic stimuli, testing generalization beyond common training distributions. Extensive experiments on both open-source and closed-source models reveal substantial limitations in current Omni-MLLMs. Based on these findings, we present a **four-level AVI taxonomy**. Overall, AVI-Bench provides a principled evaluation framework to guide the development of more robust and generalizable AVI. Project website: `https://fudancvl.github.io/AVI-Bench/`

[1]Institute of Big Data, College of Computer Science and Artificial Intelligence, Fudan University, Shanghai, China [2]Huazhong University of Science and Technology [3]Shanghai Jiaotong University [4]Renmin University of China [5]Nanyang Technological University [6]Institute for AI Industry Research (AIR), Tsinghua University [7]University of Edinburgh [8]Tsinghua University [9]University of Oxford. Correspondence to: Henghui Ding <hhding@fudan.edu.cn>.

*Proceedings of the 43rd International Conference on Machine Learning*, Seoul, South Korea. PMLR 306, 2026. Copyright 2026 by the author(s).

## 1. Introduction

The pursuit of Artificial General Intelligence (AGI) (Goertzel, 2014; Goertzel & Pennachin, 2007) has witnessed new momentum with the recent rise of Multimodal Large Language Models (MLLMs) (Fei et al., 2022; Bubeck et al., 2023; OpenAI et al., 2024; Hurst et al., 2024; Wang et al., 2025a), which leverage powerful Large Language Models (LLMs) as central reasoning engines across diverse sensory inputs. While traditional MLLMs can process individual non-linguistic modalities, such as vision, audio, or tactile input, they fall short of human-like capabilities at seamlessly integrating multiple sensory inputs to support coherent, robust, and contextually rich cross-modal reasoning.

This gap has led to the development of Omni-Multimodal Large Language Models (Omni-MLLMs), which can jointly process text, visual, and audio modalities, thereby covering the majority of human perceptual inputs. These models mark a critical step toward human-like *audio-visual intelligence (AVI)* via cross-modal perception and reasoning (Wang et al., 2025b; Li et al., 2025b; Fei et al., 2025). OpenAI's GPT-4o notably exemplifies this evolution, demonstrating sophisticated cross-modal capabilities and positioning Omni-MLLMs as promising candidates to drive the next stage of AGI-oriented research.

However, meaningful progress in Omni-MLLMs demands rigorous, structured benchmarks that can holistically evaluate cross-modal capabilities. Existing benchmarks are often modality-specific, such as MMMU (Yue et al., 2024) and SEED (Li et al., 2023) for vision-language tasks, or MMAU (Sakshi et al., 2024) for audio-language tasks, but they fail to reflect the multifaceted nature of real-world cross-modal scenarios. To this end, recent efforts have been incorporating multiple non-linguistic modalities alongside language, with a focus on tasks such as audio-visual question answering (Sun et al., 2024; Yang et al., 2022; Zhang et al., 2024), captioning (Liu et al., 2024; Sudarsanam et al., 2025), segmentation (Zhou et al., 2022; Wang et al., 2024d; 2026), and hallucination detection (Sung-Bin et al., 2024; Chowdhury et al., 2025) for cross-modal comprehension assessments, thus representing important steps toward more comprehensive evaluations of AVI in Omni-MLLMs.

Despite these advances, existing audio-visual benchmarks remain limited. Although efforts such as Omni-Bench (Li et al., 2025c), DailyOmni (Zhou et al., 2025) and AV-Odyssey (Gong et al., 2024) expand task diversity, they lack a unified and structured framework for evaluating multifaceted AVI, resulting in fragmented evidence of model capabilities and limited insight into their alignment with human-like audio-visual processing. Moreover, strong performance on isolated tasks is insufficient to indicate progress toward general intelligence, highlighting the need for cognitively aligned evaluations that reflect how humans perceive, integrate, and reason across modalities and support robust assessment in real-world scenarios.

To bridge this gap and delineate the boundaries of AVI in Omni-MLLMs, we propose the Human-like Audio-Visual Intelligence Benchmark (**AVI-Bench**), a cognitively inspired evaluation framework that systematically assesses Omni-MLLMs across three tightly integrated stages: perception, understanding, and reasoning. Each stage targets a distinct AVI aspect and comprises tasks that demand simultaneous processing and interpretation of both visual and audio inputs. This design captures the structural complexity and diversity inherent to human-like AVI within a unified evaluation framework. To further examine model

adaptation beyond commonly used general-domain training data, we introduce **AVI-Bench-PriSe**, an extension to assess whether these models exhibit "**Pri**mitive **Se**nsation" when exposed to unfamiliar and low-semantic stimuli. In addition, building on insights gained from AVI-Bench, we propose a four-level taxonomy to categorize the current AVI landscape, providing in-depth guidance for the advancing Omni-MLLMs across task-adaptive, modality-adaptive, stage-adaptive and domain-adaptive dimensions. Together, these attributes establish a rigorous and structured foundation for precise and comparative evaluations of Omni-MLLMs' human-like audio-visual intelligence.

In summary, our key contributions are:

- We introduce **AVI-Bench**, a cognitively inspired benchmark that spans the stages of perception, understanding, and reasoning with cross-modal tasks, alongside AVI-Bench-PriSe, an extension designed to assess adaptation to unfamiliar-domain inputs beyond commonly used general-domain training data.

- We conduct comprehensive evaluations on both open- and closed-source Omni-MLLMs, revealing **key challenges** that hinder progress toward robust and general AVI.

- Building on evaluation observations, we propose a **four-level principled taxonomy** for classifying the AVI of Omni-MLLMs, offering a structured and interpretable view of the current landscape.

## 2. Related works

### 2.1. Multimodal Large Language Models

LLMs such as ChatGPT (Brown et al., 2020; OpenAI et al., 2024), LLaMA (Touvron et al., 2023a;b; Grattafiori et al., 2024), and the Qwen series (Bai et al., 2023; Yang et al., 2024) have demonstrated strong capabilities in complex linguistic tasks (Wang et al., 2019; Cobbe et al., 2021; Chen et al., 2021b; Chiang et al., 2024). Building on these advances, recent studies extend LLMs to multimodal by incorporating non-linguistic modalities, including images, audio, and video. Early efforts primarily focused on coupling language with a single additional modality. Vision-Language Models (Liu et al., 2023; Zhu et al., 2023; Wang et al., 2024a; OpenAI, 2023) and Audio-Language Models (Chu et al., 2024; Zhang et al., 2023a; Deshmukh et al., 2023; Ding et al., 2025) integrate language with visual or audio inputs and achieve strong performance on tasks such as captioning and question answering. However, these approaches largely address isolated modality pairs, which limits their capacity for joint multimodal reasoning. Motivated by progress toward AGI, Omni-MLLMs aim to unify multiple sensory modalities, particularly vision and audio, within a single framework that more closely resembles human per-

ception. Early models such as PandaGPT (Su et al., 2023), VAST (Chen et al., 2023), NExT-GPT (Wu et al., 2024), AnyGPT (Zhan et al., 2024), and VideoLLaMA (Zhang et al., 2023b) established this direction by enabling cross-modal interaction. Substantial advances were later achieved by Gemini-1.5 (Team et al., 2024) and GPT4o (Hurst et al., 2024), which demonstrated robust understanding across visual and auditory modalities. More recently, a new generation of Omni-MLLMs, including Human-Omni (Zhao et al., 2025b), Baichuan-Omni-1.5 (Li et al., 2024), and Qwen2.5-Omni (Xu et al., 2025), has further improved audio-visual understanding and cross-modal alignment. Overall, the rapid evolution of Omni-MLLMs represents a key step toward AGI. By jointly modeling language, vision, and audio, these systems exhibit increasingly strong cross-modal reasoning capabilities, narrowing the gap between artificial intelligence and human-like multimodal cognition.

### 2.2. Benchmarking Omni-MLLMs

The rapid advancement of Omni-MLLMs has motivated the development of benchmarks to evaluate their multimodal capabilities, particularly in vision and audio, the two dominant non-linguistic modalities. Early evaluations primarily relied on modality-specific benchmarks, assessing vision-language or audio-language understanding in isolation. While effective for targeted analysis, these benchmarks fail to capture the integrated and synergistic nature of human perception. To overcome these limitations, recent benchmarks increasingly incorporate audio-visual tasks to assess cross-modal alignment and comprehension. Early efforts include AVQA (Yang et al., 2022) and Music-AVQA (li et al., 2022), which evaluate multimodal understanding via audio-visual question answering, and AVHBench (Sung-Bin et al., 2024) and AVTrustBench (Chowdhury et al., 2025), which examine hallucination phenomena under multimodal conditions. SAVEBench (Sun et al., 2024) extends evaluation to both unimodal and cross-modal tasks, though its coverage of audio-visual comprehension remains limited. More recent benchmarks, including AV-Odyssey (Gong et al., 2024), OmniBench (Li et al., 2025c), and OmnixR (Chen et al., 2024), further diversify tasks, domains, and modalities, capturing a broader range of real-world audio-visual scenarios. Worldsense (Hong et al., 2025), Daily-Omni (Zhou et al., 2025), and Video-Holmes (Cheng et al., 2025) evaluate Omni-MLLMs in real-world video understanding, each with a distinct focus: Worldsense emphasizes simultaneous audio-visual comprehension, Daily-Omni assesses reasoning over longer videos with attention to temporal alignment, and Video-Holmes tests complex video reasoning primarily through visual content, with audio providing supplementary context. Despite these advances, existing benchmarks have notable limitations. Most focus on task diversity without systematically assessing the development

of AVI across tasks, which hinders the diagnosis of failure modes and identification of reasoning deficiencies. Moreover, crucial capabilities, such as audio-visual grounding for localizing sound-emitting objects (Tian et al., 2018; Zhou et al., 2022; Wang et al., 2024b; Guo et al., 2025) and identifying language-referenced entities in audio-visual scenes (Wang et al., 2024d;c), are often neglected. These grounding tasks are essential for evaluating both perception and reasoning within spatially grounded contexts. In contrast, our work introduces a structured, cognitively grounded benchmark that emphasizes both breadth and systematic evaluation. By aligning tasks with distinct stages of human-like cognition, it enables principled assessment of cross-modal intelligence and supports deeper understanding of Omni-MLLM capacities and limitations.

## 3. Audio-Visual Intelligence Benchmark

### 3.1. Benchmark Overview

As shown in Table 1, compared to other benchmarks, AVI-Bench stands out for its comprehensive coverage of 9 metrics for 14 diverse tasks. These include complex tasks such as multi-instance recognition and counting, spatio-temporal localization, and text-visual-audio grounding, which are overlooked by existing benchmarks that primarily focus on simpler question types such as Multiple Choice Questions (MCQ), Yes-No, or Open-ended responses for question-answering (QA) and captioning. AVI-Bench organizes its evaluation into three stages that mirror human cognitive processes *Perception*, *Understanding*, and *Reasoning*, plus a unique stage evaluating the *Primitive Sensation* of Omni-MLLMs on unfamiliar low-semantic audio-visual inputs. By providing a structured and comprehensive evaluation, AVI-Bench offers deeper insights into AVI in Omni-MLLMs across a wide spectrum of capabilities. Our design not only follows a staged structure but also maintains a balance between visual and audio tasks at each stage. Specifically, each stage includes audio-dominant tasks (e.g., AMIC, VAR, AVH, ASQA), visual-dominant tasks (e.g., VMIC, AVR, VAH, VSQA), and tasks requiring substantial audio-visual collaboration. This organization enables a thorough assessment of the model's performance across modalities, ensuring a well-rounded evaluation at every stage. Task descriptions are provided in detail in Section 3.2.

### 3.2. Stages and Tasks: Motivation and Definition

This section presents an overview of the evaluation stages and corresponding tasks in AVI-Bench, each designed to assess distinct dimensions of AVI in Omni-MLLMs. Representative data samples for each task are shown in Figure 2.

**Perception:** The perception stage focuses on evaluating the model's ability to detect and recognize fundamental semantic entities in unimodal and multimodal inputs. This includes

*Table 1.* Comparison of key statistics across leading audio-visual benchmarks. AVI-Bench comprises 5,864 samples spanning 14 diverse tasks, evaluated using 9 metrics across three cognitively grounded stages: perception, understanding, and reasoning, along with the AVI-Bench-PriSe extension for unfamiliar-domain adaptation evaluation. AVI-Bench also features a broader range of question types such as cross-modal grounding tasks that are often overlooked in prior works and evaluations.

| Dataset | | Qualitative | | | | Quantitative | | | |
|---|---|---|---|---|---|---|---|---|---|
| | Pub. | Modality | Annotation | Answer | Grounding | #Task | #Sample | #Metric | #Stage |
| VALOR | [TPAMI'24] | T,A,V | New | Open | ✗ | 2 | 3,500 | 7 | 1 |
| SAVEBench | [ICML'24] | T,A,V,I | Repurposed | MCQ/Open/Yes-No | ✗ | 6 | 11,908 | 5 | 1 |
| WorldQA | ['24] | T,A,V | New | MCQ/Open | ✗ | 1 | 1,007 | 2 | 1 |
| AVCaps | [OJSP'24] | T,A,V | New | Open | ✗ | 2 | 2,061 | 8 | 1 |
| AV-Odyssey | ['24] | T,A,V,I | New | MCQ | ✗ | 7 | 4,555 | 2 | 1 |
| DailyOmni | ['25] | T,A,V | New | MCQ | ✗ | 6 | 1,197 | 1 | 1 |
| AVTrustBench | [ICCV'25] | T,A,V | Repurposed | MCQ | ✗ | 9 | 600,000 | 1 | 1 |
| OmniBench | [NeurIPS'25] | T,A,I | New | MCQ | ✗ | 8 | 1,142 | 1 | 1 |
| OmnixR | [ICLR'25] | T,A,V,I | Hybrid | MCQ | ✗ | 6 | - | 1 | 1 |
| AVHBench | [ICLR'25] | T,A,V | Hybrid | MCQ/Open | ✗ | 4 | 5,302 | 7 | 1 |
| AVI-Bench | [ICML'26] | T,A,V,I | Hybrid | MCQ/Open/Yes-No BBox/Number/List | ✓ | 14 | 5,864 | 9 | 4 |

identifying salient objects, events, or sources in either the audio or visual stream, as well as aligning information across modalities at both local and global levels. As illustrated at the top of Figure 2, Audio Multi-instance Classification (AMIC) (Lee et al., 2009; Zaman et al., 2023) and Visual Multi-instance Classification (VMIC) (Naeem et al., 2023; Pratt et al., 2023) assess unimodal perception capabilities by requiring the detection of multiple co-occurring audio or visual instances within a single sample. To further evaluate cross-modal alignment, we include Audio-Visual Localization (AVL) (Chen et al., 2021a; Mo & Morgado, 2022; Zhou et al., 2022), which requires identifying the spatial location of sound sources within a visual scene, and Audio-Visual Matching (AVM) (Lee et al., 2022; Sung-Bin et al., 2024), which assesses the ability to determine whether audio-visual inputs correspond globally. Together, these tasks provide a foundation for measuring models' capacity to perceive and align multimodal information at a fine-grained level.

**Understanding:** The understanding stage evaluates a model's ability to integrate and reason over multimodal context, which is critical for interpreting real-world scenes with temporal and semantic dependencies. Audio-Visual Captioning (AVC) (Liu et al., 2024; Sudarsanam et al., 2025) measures narrative understanding by assessing the generation of coherent, context-aware descriptions from audio and visual inputs. In addition, cross-modal retrieval tasks, including Audio-reference Visual Retrieval (AVR) and Visual-reference Audio Retrieval (VAR) (Zhang et al., 2023c; Liu et al., 2024; Sudarsanam et al., 2025), evaluate cross-modal association by requiring temporal and semantic alignment between audio and visual content.

**Reasoning:** The reasoning stage probes the model's ability to perform higher-order inference over integrated mul-

timodal and linguistic information. This stage moves beyond recognition and contextual understanding, requiring the model to synthesize information, draw conclusions, and make judgments based on complex semantic relationships across audio and visual modalities. Specifically, Audio-Visual Question Answering (AVQA) (Yang et al., 2022; Yun et al., 2021; li et al., 2022) targets coarse-grained reasoning by requiring the model to answer questions that rely on a holistic understanding of audio-visual events. In contrast, Audio-Visual Language Grounding (AVLG) (Wang et al., 2024c;d) focuses on fine-grained reasoning, requiring precise localization of objects or events referenced in natural language. To further evaluate model robustness under ambiguous or conflicting input conditions, we include Audio-reference Visual Hallucination (AVH) and Visual-reference Audio Hallucination (VAH) (Sung-Bin et al., 2024; Chowdhury et al., 2025). These tasks assess the model's susceptibility to hallucination when exposed to cross-modal inconsistencies, providing insights into its resilience and reliability in complex, noisy environments.

**Primitive Sensation:** While most existing Omni-MLLMs are trained on large-scale, curated datasets rich in semantic content, it remains unclear whether they can adapt beyond such commonly used distributions to exhibit human-like perceptual sensitivity. This raises a fundamental question: Can these models perform low-level sensory tasks that are trivially easy for humans, such as detecting variations in color, volume, texture, or geometry, especially when semantic context is minimal or absent? To this end, we introduce AVI-Bench-PriSe, a supplementary suite evaluating the primitive sensation capabilities of Omni-MLLMs. It focuses on the model's response to naive, unfamiliar, and low-semantic audio-visual inputs beyond conventional training data. As shown at the bottom of Figure 2, this stage

includes three tasks: Audio Sensation Question Answering (ASQA), Visual Sensation Question Answering (VSQA), and Audio-Visual Sensation Question Answering (AVSQA). These tasks use controlled and low-semantic data to investigate the difference between authentic human-like AVI and mere pattern fitting. This stage provides a new lens to evaluate the fundamental sensory of Omni-MLLMs and their limitations in replicating core aspects of human-like AVI.

### 3.3. Task Sample Counts

Table 2 further demonstrates the detailed sample counts across different tasks and stages. In AVI-Bench, 62% of the data consists of fully manually constructed samples, covering tasks such as AMIC, VMIC, VAR, AVR, ASQA, VSQA, and AVSQA, totaling 3,657 samples. Additionally, some tasks involve converting dense mask annotations into bounding boxes with normalized width and height, such as AVL and AVLG, yielding 708 samples. Other tasks restructure existing data into a unified JSON format while preserving the original content, including AVM, AVC, AVH, VAH, and AVQA, comprising 1,499 samples.

## 4. Experiments

### 4.1. Models

AVI-Bench conducts a comprehensive evaluation of 28 Omni-MLLMs with audio-visual capabilities. The evaluation encompasses both closed-source models such as GPT-4o (Brown et al., 2020) and the Gemini series (Team et al., 2024), and open-source counterparts, including Qwen-2.5-Omni (Xu et al., 2025), Ola (Liu et al., 2025), and Baichuan-Omni-1.5 (Li et al., 2025a). While most evaluated models have over 7 billion parameters, we also include a set of smaller models, such as Human-Omni-0.5B (Zhao et al., 2025b), R1-Omni-0.5B (Zhao et al., 2025a), and Phi-4-Multimodal (Abouelenin et al., 2025), to investigate performance trends across different model scales.

### 4.2. Results Analysis and Observations

Our evaluation of 28 Omni-MLLMs on AVI-Bench provides several critical insights into the current capabilities of Omni-MLLMs and their potential to achieve human-like AVI:

**Observation 1: Synergy Across Cognitive Stages.** As shown in Figure 3, darker red regions in the heatmap, indicating higher performance, are concentrated on the left. This pattern reflects consistently strong scores across perception, understanding, and reasoning, suggesting positive correlations among these stages. Models that excel in reasoning also tend to perform well in perception and understanding, highlighting the interconnected and synergistic nature of cognitive skills in achieving comprehensive AVI.

**Observation 2: Perception and Understanding Limit Reasoning.** Beyond cross-stage synergy, perception and understanding critically constrain reasoning, often creating a bottleneck. Insufficient performance in either domain leads to degraded reasoning. Figure 3 shows several open-source models exhibit imbalanced capabilities: Baichuan-Omni-1.5, PandaGPT-7B, and PandaGPT-13B demonstrate strong understanding but weak perception, whereas Video-LLaMA2 shows in opposite. In both cases, reasoning remains limited, highlighting the need to improve both perception and understanding to advance cross-modal reasoning.

**Observation 3: Imbalance Between Audio and Visual Intelligence.** As shown in Table 3, most Omni-MLLMs excel on visual-dominant tasks (e.g., VSQA, VMIC, AVR, AVH) but lag on audio-dominant ones, revealing audio intelligence as a key bottleneck and highlighting substantial room for improvement in these models' audio processing.

**Observation 4: Model Scale Correlates with Performance.** As expected, larger models consistently outperform their smaller counterparts across nearly all tasks. For example, PandaGPT-13B surpasses PandaGPT-7B, and Human-Omni-7B significantly outperforms Human-Omni-0.5B. Interestingly, Phi-4-Multimodal, which employs a mixture-of-LoRA approach, outperforms several 7B models, suggesting that there is substantial room to optimize AVI even within models with more modest parameters. This highlights the importance of model architecture and training strategies in advancing multimodal capabilities.

**Observation 5: Grounding Remains a Persistent Challenge.** As depicted in Table 3, tasks requiring grounding capabilities, such as AVL and AVLG, remain highly challenging. Even the top-performing model, Gemini-2.5-Pro, achieves only 39.1% on AVL and 35.1% on AVLG. Notably, no open-source model surpasses 25.0% in either AVL or AVLG task. These results emphasize that fine-grained audio-visual grounding remains a major challenge for current Omni-MLLMs, particularly for open-source ones.

**Observation 6: Unfamiliar Domain Adaptation Remains a Challenge.** Our comparison of performance on reasoning and primitive sensation tasks highlights the major challenges faced by current Omni-MLLMs. These tasks assess model robustness using low-semantic inputs, which differ from the conventional training data. As shown in Table 3, although the reasoning stage includes highly challenging tasks like AVLG, which lower the overall stage score, no model showed superior performance in primitive sensation compared to reasoning tasks. The relative performance gap between reasoning and primitive sensation ranges from 47.6% (Gemini-2.5-Pro) to 82.7% (Video-Salmonn), highlighting the difficulty these models encounter when handling low-semantic data, which differs significantly from the commonly used training data. This underscores the need for fur-

*Table 2.* Per-task sample counts and modality balance across the four stages of AVI-Bench. AVI-Bench organizes 14 tasks into the categories of Perception, Understanding, Reasoning, and Primitive Sensation. Within each stage, we deliberately balance audio-dominant tasks (AMIC, VAR, AVH, ASQA), visual-dominant tasks (VMIC, AVR, VAH, VSQA), and collaboration-intensive tasks to ensure that no stage score is dominated by a single modality.

| Stage 1: Perception | Count | Stage 2: Understand | Count | Stage 3: Reasoning | Count | Stage 4: Primitive Sensation | Count |
|---|---|---|---|---|---|---|---|
| AMIC | 518 | VAR | 264 | AVH | 250 | ASQA | 502 |
| VMIC | 521 | AVR | 264 | VAH | 250 | VSQA-img | 620 |
| AVL | 205 | AVC | 280 | AVQA | 469 | VSQA-vid | 580 |
| AVM | 250 | | | AVLG | 503 | AVSQA | 388 |
| # | 1494 | # | 808 | # | 1472 | # | 2090 |

*Table 3.* Evaluation results of AVI-Bench across 28 Omni-MLLMs. All task scores are normalized to percentages for unified comparison, with higher values indicating better performance.

| Omni-MLLMs | Params. | Perception | | | | | Understand | | | | Reasoning | | | | | Primitive Sensation | | | | avg. |
|---|---|---|---|---|---|---|---|---|---|---|---|---|---|---|---|---|---|---|---|---|
| | | AMIC | VMIC | AVL | AVM | avg. | VAR | AVR | AVC | avg. | AVH | VAH | AVQA | AVLG | avg. | ASQA | VSQA | AVSQA | avg. | |
| Gemini-2.5-pro | - | 43.01 | 59.39 | 39.13 | 76.80 | 54.58 | 76.04 | 74.42 | 56.46 | 68.97 | 86.40 | 80.80 | 73.93 | 35.08 | 69.06 | 29.48 | 62.67 | 16.50 | 36.22 | 57.21 |
| Gemini-2.5-flash | - | 27.71 | 55.78 | 39.18 | 61.20 | 45.97 | 38.83 | 34.57 | 57.97 | 43.79 | 79.20 | 70.80 | 72.01 | 32.79 | 63.70 | 23.11 | 44.04 | 24.74 | 30.63 | 46.02 |
| Gemini-2.0-flash | - | 24.91 | 48.62 | 37.93 | 65.60 | 44.27 | 39.26 | 30.75 | 56.31 | 42.11 | 84.80 | 75.20 | 68.51 | 27.61 | 64.03 | 21.51 | 40.13 | 26.80 | 29.48 | 44.97 |
| Qwen2.5-Omni | 7B | 32.87 | 40.60 | 19.36 | 78.40 | 42.81 | 35.49 | 26.32 | 57.22 | 39.68 | 82.40 | 77.60 | 64.29 | 08.74 | 58.26 | 21.12 | 31.51 | 21.13 | 24.59 | 41.33 |
| Qwen-Omni-turbo | 7B | 32.87 | 40.62 | 19.21 | 78.40 | 42.77 | 36.39 | 26.24 | 57.01 | 39.88 | 80.40 | 74.40 | 63.71 | 08.86 | 56.84 | 21.51 | 33.40 | 21.65 | 25.52 | 41.25 |
| GPT-4o | - | 24.18 | 43.35 | 20.66 | 73.60 | 40.45 | 39.70 | 52.69 | 53.40 | 48.60 | 87.60 | 64.80 | 54.03 | 21.07 | 56.87 | 00.40 | 40.27 | 09.75 | 16.81 | 40.68 |
| Gemini-1.5-pro | - | 12.96 | 53.35 | 43.63 | 56.80 | 41.69 | 25.17 | 25.78 | 55.03 | 35.33 | 85.20 | 62.80 | 67.23 | 29.54 | 61.19 | 17.93 | 28.27 | 23.71 | 23.30 | 40.38 |
| Gemini-1.5-flash | - | 10.56 | 46.79 | 36.33 | 54.40 | 37.02 | 16.80 | 27.72 | 54.00 | 32.84 | 84.80 | 66.40 | 69.80 | 30.29 | 62.82 | 21.91 | 24.06 | 17.01 | 20.99 | 38.42 |
| Ola | 7B | 30.54 | 44.90 | 17.88 | 57.20 | 37.63 | 28.02 | 09.50 | 39.67 | 25.73 | 79.20 | 83.20 | 58.21 | 07.07 | 56.92 | 11.95 | 27.47 | 11.86 | 17.09 | 34.34 |
| Baichuan-Omni | 7B | 22.59 | 20.96 | 12.89 | 59.20 | 28.91 | 28.14 | 39.42 | 41.83 | 36.46 | 74.00 | 54.40 | 55.59 | 02.75 | 46.69 | 24.70 | 23.56 | 13.40 | 20.55 | 33.15 |
| GPT-4o-mini | - | 19.54 | 32.72 | 18.49 | 52.00 | 30.69 | 31.55 | 32.95 | 52.25 | 38.92 | 74.80 | 34.80 | 50.39 | 17.47 | 44.37 | 00.00 | 25.85 | 06.19 | 10.68 | 31.16 |
| Reka-flash | 21B | 15.36 | 39.98 | 17.28 | 48.40 | 30.26 | 17.08 | 23.40 | 48.30 | 29.59 | 66.80 | 48.40 | 57.55 | 20.55 | 48.33 | 21.91 | 16.88 | 01.55 | 13.45 | 30.41 |
| Phi-4-Multimodal | 5.6B | 07.35 | 39.20 | 10.61 | 52.40 | 27.39 | 03.69 | 23.74 | 51.04 | 26.16 | 86.80 | 60.40 | 53.75 | 02.27 | 50.80 | 01.59 | 33.26 | 00.52 | 11.79 | 29.04 |
| Human-Omni | 7B | 26.54 | 34.31 | 01.96 | 52.00 | 28.70 | 01.78 | 06.56 | 46.08 | 18.14 | 86.00 | 57.20 | 56.54 | 00.02 | 49.94 | 15.54 | 19.47 | 15.46 | 16.82 | 28.40 |
| Ixc2.5-OL | 7B | 08.37 | 47.86 | 05.68 | 52.40 | 28.58 | 06.80 | 11.20 | 51.61 | 23.20 | 80.40 | 58.40 | 53.23 | 13.27 | 51.33 | 04.38 | 25.43 | 01.03 | 10.28 | 28.35 |
| Video-LLaMA2 | 7B | 36.46 | 38.25 | 02.68 | 52.40 | 32.45 | 00.00 | 02.80 | 46.17 | 16.32 | 44.80 | 70.00 | 44.16 | 00.86 | 39.95 | 24.30 | 15.06 | 21.65 | 20.34 | 27.27 |
| OneLLM | 7B | 09.55 | 38.67 | 02.30 | 52.00 | 25.63 | 00.00 | 07.29 | 45.30 | 17.53 | 79.60 | 60.80 | 40.59 | 00.00 | 45.25 | 24.70 | 10.85 | 19.07 | 18.21 | 26.65 |
| VITA-1.5 | 7B | 05.70 | 38.55 | 12.54 | 45.20 | 25.50 | 04.79 | 19.68 | 36.08 | 20.18 | 84.80 | 54.00 | 34.11 | 01.45 | 41.89 | 00.40 | 31.74 | 03.61 | 11.92 | 24.87 |
| PandaGPT | 13B | 17.30 | 26.19 | 01.34 | 42.40 | 21.81 | 23.70 | 21.85 | 40.66 | 28.74 | 38.80 | 54.00 | 28.22 | 00.08 | 30.27 | 13.15 | 15.55 | 07.73 | 12.14 | 23.24 |
| Video-salmonn | 13B | 13.22 | 38.45 | 02.04 | 53.20 | 26.73 | 04.70 | 03.74 | 48.15 | 18.86 | 57.60 | 58.00 | 42.73 | 01.38 | 39.93 | 07.57 | 06.48 | 06.70 | 06.92 | 23.11 |
| PandaGPT | 7B | 04.77 | 17.68 | 01.46 | 42.00 | 16.48 | 24.92 | 25.69 | 41.28 | 30.63 | 43.20 | 56.00 | 35.12 | 00.08 | 33.60 | 10.76 | 08.59 | 07.73 | 09.03 | 22.43 |
| UniMoE | 7Bx4 | 00.00 | 22.09 | 01.42 | 46.80 | 17.58 | 09.65 | 07.28 | 07.64 | 08.19 | 59.60 | 50.80 | 34.43 | 07.36 | 38.05 | 18.33 | 14.77 | 05.67 | 12.92 | 19.18 |
| Imagebind-LLM | 7B | 17.30 | 18.94 | 01.42 | 41.60 | 19.82 | 11.23 | 12.28 | 38.86 | 20.79 | 44.40 | 50.40 | 31.11 | 00.02 | 30.48 | 03.98 | 01.29 | 00.52 | 01.93 | 18.25 |
| X-instruct-BLIP | 7B | 00.00 | 14.73 | 01.37 | 52.40 | 17.12 | 05.18 | 07.00 | 47.20 | 19.79 | 38.40 | 15.20 | 36.07 | 00.00 | 22.42 | 00.00 | 06.14 | 12.37 | 06.17 | 16.38 |
| Mergrez-Omni | 3B | 00.00 | 29.09 | 01.43 | 28.40 | 14.73 | 07.59 | 03.62 | 38.91 | 16.71 | 38.80 | 26.80 | 40.44 | 00.27 | 26.58 | 00.00 | 22.07 | 00.00 | 07.36 | 16.34 |
| Human-Omni | 0.5B | 00.00 | 03.77 | 01.28 | 04.00 | 02.26 | 00.77 | 05.52 | 35.59 | 13.96 | 36.80 | 43.60 | 37.08 | 00.12 | 29.40 | 00.00 | 04.56 | 14.43 | 06.33 | 12.99 |
| NExTGPT | 7B | 00.00 | 00.00 | 01.28 | 28.80 | 07.52 | 00.00 | 13.86 | 36.92 | 16.93 | 11.60 | 29.60 | 16.75 | 01.47 | 14.86 | 11.95 | 12.40 | 07.73 | 10.69 | 12.50 |
| R1-Omni | 0.5B | 00.00 | 00.00 | 05.01 | 18.80 | 05.95 | 04.32 | 06.15 | 30.89 | 13.79 | 18.80 | 20.00 | 07.36 | 00.00 | 11.54 | 00.00 | 03.35 | 00.00 | 01.12 | 08.10 |

ther research to enhance the adaptability of Omni-MLLMs and enable them to achieve true human-like AVI.

### 4.3. Human Performance

While all dataset samples were manually verified, assuming ideal human performance, it remains important to evaluate multiple human subjects to mitigate individual variability and obtain a reliable estimate of average human performance. To this end, we conducted a pilot study with six participants on a subset of tasks and compared their performance with the top-performing model, *Gemini-2.5-Pro*. Participants were allowed to freely replay audio and video stimuli before submitting their responses. As reported in Table 4, human participants perform consistently well across all cognitive levels. The largest performance gap occurs at the sensation level, particularly on AVSQA, where *Gemini-2.5-Pro* achieves only 16.50 compared to the human score of 90.55. These results highlight both the difficulty of the benchmark and the substantial gap that Omni-MLLMs must overcome to achieve human-level audio-visual intelligence.

However, "human-like" does not imply direct human-model performance comparisons, which can be misleading given that recent multimodal and language models surpass humans on many benchmarks. Instead, we advocate a *methodological evaluation* of Omni-MLLMs from a human cognitive perspective via our four-level framework: task-, modality-, stage-, and domain-adaptive evaluation.

## 5. Classifying Human-like AVI

As Omni-MLLMs advance in handling complex audio-visual tasks, there remains a lack of structured criteria to quantify how closely their capabilities approximate human-like intelligence. While absolute task performance provides

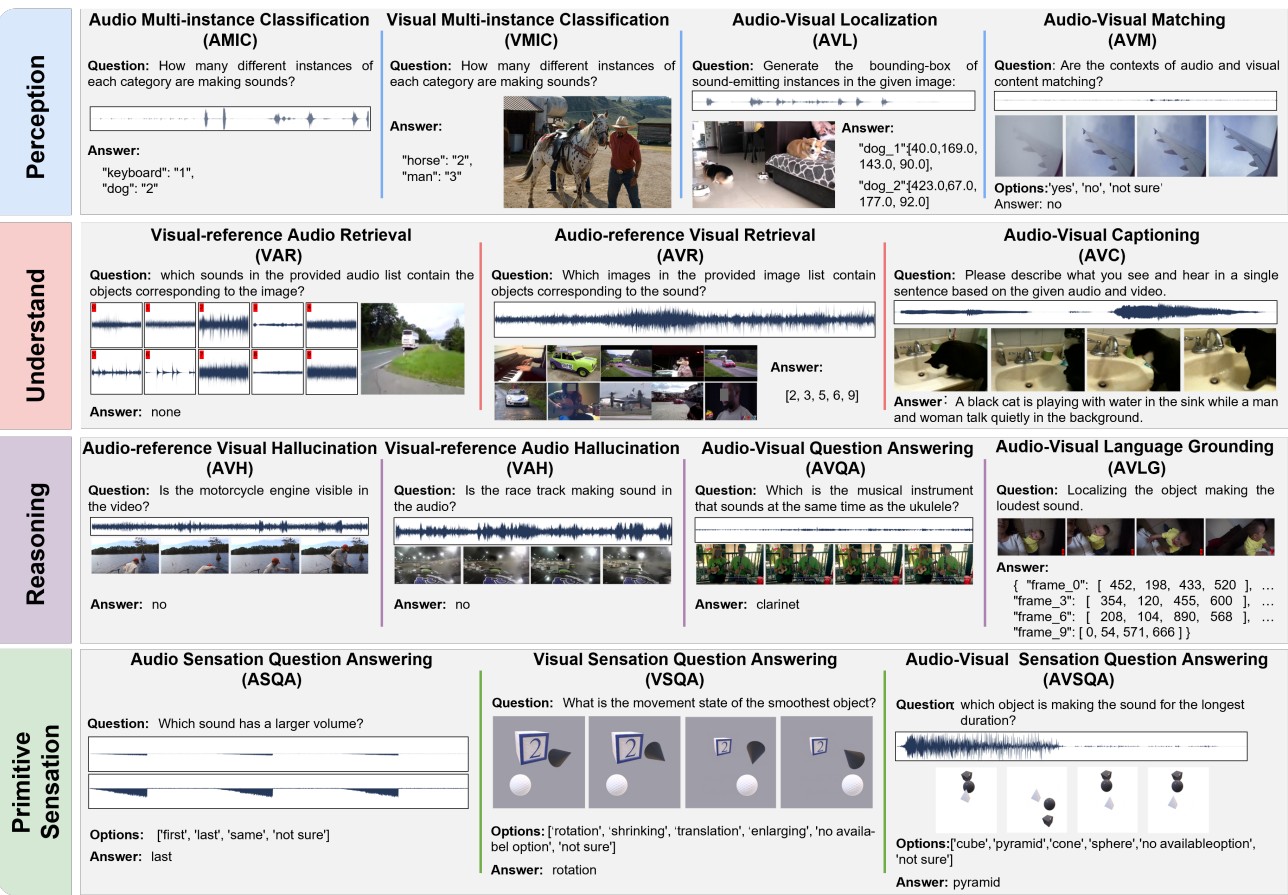

*Figure 2.* Data samples spanning the three cognitively inspired stages of AVI-Bench: perception, understanding, and reasoning. Furthermore, we introduce AVI-Bench-PriSe, an extension aim at evaluating whether Omni-MLLMs exhibit human-like audio-visual capabilities by adapting to unfamiliar and low-semantic data.

*Table 4.* Comparison between human performance and *Gemini-2.5-Pro* across cognitive levels and tasks.

| Cognitive | Task | Gemini-2.5-Pro | Human |
|---|---|---|---|
| Perception | AMIC | 43.01 | 86.34 |
| | VMIC | 59.39 | 94.82 |
| | AVM | 76.80 | 95.29 |
| Understand | VAR | 76.04 | 90.47 |
| | AVR | 74.42 | 89.63 |
| Reasoning | AVH | 86.40 | 97.00 |
| | VAH | 80.80 | 99.50 |
| | AVLG | 35.08 | 96.48 |
| Sensation | ASQA | 29.48 | 86.11 |
| | VSQA | 62.67 | 92.30 |
| | AVSQA | 16.50 | 90.55 |

a partial view, it often overlooks crucial aspects like cross-modal balance, cognitive-stage synergy, and adaptation to unfamiliar domains. To address this, we propose a hierarchical four-level classification scheme to characterize AVI in Omni-MLLMs. Each level represents a progressively stricter and more human-aligned criterion, enabling systematic and interpretable assessments of model capabilities beyond surface-level task accuracy.

## 5.1. Level-1: Task-Adaptive Intelligence

The first and most fundamental level, *task-adaptive intelligence*, denotes a model's ability to achieve consistent performance across a wide range of audio-visual tasks. This level establishes the baseline competency expected of any Omni-MLLM. Specifically, given a set of tasks $\mathcal{T} = \{t_1, t_2, \ldots, t_n\}$, where each task $t_i$ has an associated performance metric $\mathcal{F}(\cdot)$, the task-adaptive score $\mathcal{S}_T$ is defined as the average performance across all tasks:

$$\mathcal{S}_T = \frac{\sum_{t_i \in T} \mathcal{F}(t_i)}{|T|}, \tag{1}$$

where $\mathcal{F}(t_i)$ denotes the performance metric for task $t_i$, and $|\mathcal{T}|$ is the total number of tasks. $\mathcal{S}_T$ aggregates performance from various tasks to establish a baseline score, serving as the foundation for deeper and more structured evaluation in subsequent levels.

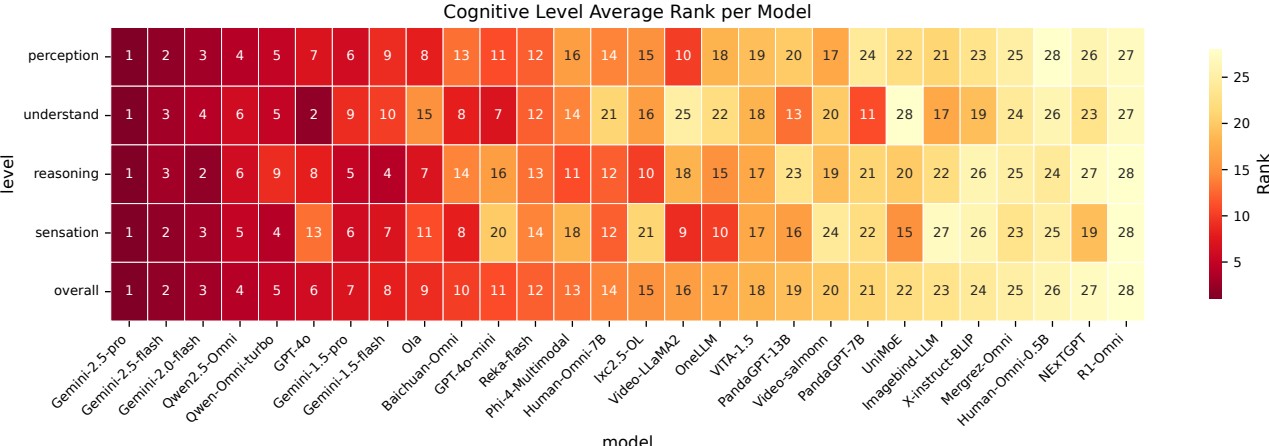

*Figure 3.* Heatmap showing the rankings of Omni-MLLMs across different stages. Darker red indicates higher rankings and stronger performance. The Gemini series consistently demonstrates strong performance throughout AVI-Bench. Among open-source models, the Qwen-2.5-Omni series also exhibits notable AVI.

## 5.2. Level-2: Modality-Adaptive Intelligence

As mentioned in Section 4.2, our Observation 3 reveals a pronounced disparity between the visual and audio intelligence of current Omni-MLLMs, with most models exhibiting dominant proficiency in visual processing, i.e., effectively acting as "visual specialists". To mitigate this imbalance, the second level, *modality-adaptive intelligence*, seeks to promote a more balanced and synergistic advancement across both visual and audio modalities. Let $\mathcal{A}$ and $\mathcal{V}$ denote the performance on audio-dominant tasks and visual-dominant tasks, respectively. The modality-specific intelligence difference $\Delta_m$ quantifies the relative discrepancy between audio and visual modalities:

$$
\Delta_m = \begin{cases} 2, & \text{if } \mathcal{A} + \mathcal{V} = 0, \\ 2 \cdot \dfrac{|\mathcal{A} - \mathcal{V}|}{\mathcal{A} + \mathcal{V}}, & \text{otherwise} \end{cases} \quad (2)
$$

then the modality-adaptive score $\mathcal{S}_M$ is calculated based on the relative discrepancy $\Delta_m$ and the fundamental task-adaptive score $\mathcal{S}_T$:

$$
\mathcal{S}_M = (1 - \alpha \cdot \Delta_m) \cdot \mathcal{S}_T, \quad (3)
$$

where the scaling constant $\alpha$ is set to $0.5$, ensuring $\mathcal{S}_T \in [0, 1]$. The modality-adaptive score $\mathcal{S}_M$ is designed to encourage both high task performance and balanced modality-specific abilities, promoting more robust AVI.

## 5.3. Level-3: Stage-Adaptive Intelligence

Based on Observation 2 outlined in Section 4.2, which highlights the bottleneck effect of perception and understanding on reasoning capability, the third level, *stage-adaptive intelligence*, measures whether reasoning is properly grounded

in its perceptual and conceptual prerequisites. Let $\mathcal{S}_P$, $\mathcal{S}_U$, and $\mathcal{S}_R$ denote the average task scores of the three cognitive stages. Because the tasks comprising different stages possess different intrinsic difficulty (e.g., binary hallucination detection has a higher random-guess floor than mIoU-based localization), we first map each raw task score to a *headroom-normalized* score that removes the task-specific chance baseline $c_t$:

$$
\tilde{m}_t = \max\left(0, \frac{m_t - c_t}{100 - c_t} \cdot 100\right), \quad (4)
$$

where $c_t$ is determined by the structure of task $t$ (e.g., $c_t = 100/k$ for a $k$-option MCQ, $c_t \approx 0$ for open-ended generation or mIoU, and the empirical FENSE floor obtained from random reference pairs for AVC). The chance baselines used in this paper are listed in Appendix G.2.2 and are independent of the evaluated model cohort. The headroom-normalized stage averages $\tilde{\mathcal{S}}_P$, $\tilde{\mathcal{S}}_U$, and $\tilde{\mathcal{S}}_R$ are then obtained by averaging $\tilde{m}_t$ within each stage.

Following Observation 2, perception and understanding are prerequisites that limit reasoning capability. We quantify the gap by which reasoning exceeds its weakest prerequisite as the *bottleneck discrepancy*:

$$
\Delta_s = \begin{cases} 0, & \text{if } \tilde{\mathcal{S}}_R \leq \min(\tilde{\mathcal{S}}_P, \tilde{\mathcal{S}}_U), \\ \dfrac{\tilde{\mathcal{S}}_R - \min(\tilde{\mathcal{S}}_P, \tilde{\mathcal{S}}_U)}{\tilde{\mathcal{S}}_R}, & \text{otherwise.} \end{cases}
$$
$$(5)$$

Intuitively, $\Delta_s = 0$ when reasoning is fully supported by perception and understanding (no bottleneck); $\Delta_s \to 1$ when reasoning runs ahead of its weakest cognitive prerequisite, indicating an ungrounded reasoning regime that the bottleneck hypothesis flags as suspicious. The stage-

*Table 5.* Comparison of 28 Omni-MLLMs under the four-level AVI taxonomy. Levels 1–4 evaluate *Task-*, *Modality-*, *Stage-*, and *Domain-Adaptive* intelligence, respectively, where higher values indicate stronger capability. Each level is normalized to a chance baseline to ensure comparability across tasks with different levels of difficulty. Models are ranked according to Level 4.

| Models | Params. | Level 1 | Level 2 | Level 3 | Level 4 | Models | Params. | Level 1 | Level 2 | Level 3 | Level 4 |
|---|---|---|---|---|---|---|---|---|---|---|---|
| Gemini-2.5-pro | - | 64.20 | 62.80 | 57.08 | 32.97 | PandaGPT | 7B | 26.90 | 22.77 | 17.71 | 11.04 |
| Gemini-2.5-flash | - | 51.15 | 48.58 | 40.47 | 27.72 | UniMoE | 7Bx4 | 21.27 | 19.72 | 09.86 | 10.63 |
| Gemini-2.0-flash | - | 50.14 | 49.21 | 39.79 | 27.12 | Video-salmonn | 13B | 28.51 | 24.50 | 18.72 | 09.52 |
| Qwen-Omni-turbo | - | 46.50 | 45.15 | 37.70 | 26.13 | NExTGPT | 7B | 13.10 | 05.52 | 05.16 | 06.92 |
| Qwen2.5-Omni | 7B | 46.92 | 45.93 | 37.61 | 25.89 | Ixc2.5-OL | 7B | 34.37 | 30.84 | 21.21 | 05.29 |
| Baichuan-Omni | 7B | 37.35 | 35.80 | 30.18 | 24.10 | Phi-4-Multimodal | 5.6B | 34.78 | 30.77 | 22.75 | 02.05 |
| Gemini-1.5-flash | - | 44.23 | 39.20 | 28.48 | 23.51 | Imagebind-LLM | 7B | 23.70 | 21.70 | 20.84 | 01.81 |
| Gemini-1.5-pro | - | 46.07 | 42.84 | 32.67 | 23.28 | VITA-1.5 | 7B | 29.19 | 24.57 | 17.56 | 00.58 |
| Video-LLaMA2 | 7B | 29.57 | 24.99 | 18.71 | 16.99 | GPT-4o | - | 48.64 | 47.19 | 41.93 | 00.55 |
| Human-Omni | 7B | 32.26 | 29.79 | 19.62 | 16.96 | GPT-4o-mini | - | 37.99 | 33.73 | 30.88 | 00.00 |
| Reka-flash | 21B | 36.06 | 33.92 | 27.23 | 16.37 | X-Instruct-BLIP | 7B | 19.78 | 18.14 | 18.14 | 00.00 |
| PandaGPT | 13B | 26.94 | 23.65 | 23.65 | 15.13 | Mergrez-Omni | 3B | 19.34 | 16.94 | 16.37 | 00.00 |
| Ola | 7B | 40.09 | 40.07 | 26.88 | 14.96 | R1-Omni | 0.5B | 10.43 | 09.79 | 09.79 | 00.00 |
| OneLLM | 7B | 29.47 | 26.82 | 18.32 | 13.84 | Human-Omni | 0.5B | 15.21 | 12.63 | 07.10 | 00.00 |

adaptive score is then

$$\mathcal{S}_S = (1 - \alpha \cdot \Delta_s) \cdot \mathcal{S}_M, \qquad \alpha = 0.5. \tag{6}$$

This formulation directly encodes the bottleneck hypothesis: a model whose reasoning ability remains anchored to its perceptual and conceptual support is rewarded, while a model achieving high reasoning without commensurate grounding incurs proportional penalty.

### 5.4. Level-4: Domain-Adaptive Intelligence

The final level, *domain-adaptive intelligence*, evaluates domain adaptation by distinguishing between familiar-domain (FD) and unfamiliar-domain (UD) performance. First, the unfamiliar-domain score, denoted as $\mathcal{S}_{UD}$, is computed using the same method described in Equation (2) and Equation (3) to compute the modality-adaptive score. Then, by defining the familiar-domain score $\mathcal{S}_{FD} = \mathcal{S}_S$, the domain-adaptive score $\mathcal{S}_D$ is computed as the harmonic mean of $\mathcal{S}_{FD}$ and $\mathcal{S}_{UD}$:

$$\mathcal{S}_D = \begin{cases} 0, & \text{if } \mathcal{S}_{FD} + \mathcal{S}_{UD} = 0, \\ 2 \cdot \frac{\mathcal{S}_{FD} \cdot \mathcal{S}_{UD}}{\mathcal{S}_{FD} + \mathcal{S}_{UD}}, & \text{otherwise.} \end{cases} \tag{7}$$

The domain-adaptive score $\mathcal{S}_D$ captures both familiar-domain performance and unfamiliar-domain adaptation, integrating priors to provide a principled indicator of human-like audio-visual intelligence.

### 5.5. Comparison of Intelligence

By comparing the model rankings in Table 3 and Table 5, we observe that the performance gaps among certain models, which appear close when evaluated using naive average scores on AVI-Bench, become significantly amplified under our proposed four-level AVI taxonomy. For example, although Gemini-1.5-pro and GPT-4o achieve similar overall performance with only a 0.30% score difference and a rank gap of 1 in Table 3, their level-wise gap progressively widens from 2.57% (L1) to 4.35% (L2), 9.26% (L3), and 22.73% (L4), with the L4 rank gap reaching 15 in Table 5. The widening reflects GPT-4o's near-zero audio sensation performance (ASQA = 0.4%) being amplified by the harmonic-mean construction of L4. This four-level taxonomy comprehensively assesses models across task adaptation, modality adaptation, cognitive stage adaptation, and domain adaptation. Consequently, it provides a more nuanced and in-depth evaluation of Omni-MLLMs' human-like AVI than simple aggregated metrics.

## 6. Conclusion

In this work, we introduced AVI-Bench, a cognitively inspired benchmark designed to comprehensively evaluate the *human-like audio-visual intelligence* of Omni-MLLMs across the stages of *perception, understanding, and reasoning*. To further assess models' adaptation beyond curated training distributions, we proposed AVI-Bench-PriSe, a supplementary testbed focused on evaluating performance under *unfamiliar and low-semantic* audio-visual stimuli. Through extensive evaluation of 28 open- and closed-source Omni-MLLMs, we identified critical limitations in current models. In particular, we observed synergistic dependencies across cognitive stages, with perception and understanding playing a critical role in supporting reasoning. In addition, we identified persistent challenges in fine-grained perception and reasoning as well as in robustness to unfamiliar domains. Building upon these insights, we proposed a *four-level taxonomy* for classifying and interpreting the development of AVI in Omni-MLLMs. We believe that our benchmark and taxonomy provide a rigorous and unified framework for future research, laying a solid foundation for pursuing human-like AVI and contributing to the broader advancement of artificial general intelligence.

## Acknowledgment

This work was supported by the National Natural Science Foundation of China (NSFC) under Grant No. 62472104 and the Science and Technology Commission of Shanghai Municipality under Grant No. 25511103600.

## Impact Statement

This paper introduces AVI-Bench, a cognitively grounded benchmark for evaluating audio-visual intelligence in Omni-Multimodal Large Language Models (Omni-MLLMs). AVI-Bench provides a structured framework to assess cross-modal perception, understanding, reasoning, and adaptation to unfamiliar domains, advancing evaluation from fragmented task-specific metrics toward holistic measurement of human-like capabilities. It enables transparent comparison of Omni-MLLMs on balanced audio-visual tasks, revealing critical bottlenecks such as modality imbalance and weak grounding capabilities. By employing a four-level taxonomy, AVI-Bench guides responsible development that considers not only task performance but also modality balance, cognitive-stage synergy, and domain adaptation, promoting research toward robust and generalizable multimodal intelligence rather than narrow task-specific optimization.

While enhanced audio-visual capabilities could potentially be misused, AVI-Bench itself poses no risk and is designed as an open evaluation resource to support safety research and model auditing. Benchmark performance alone cannot guarantee real-world safety, and AVI-Bench is intended solely as a diagnostic tool. All data were rigorously verified: familiar-domain samples were sourced from publicly licensed datasets, and unfamiliar-domain samples in AVI-Bench-PriSe were synthetically constructed offline, with no personally identifiable information included. We believe that responsible use of AVI-Bench can positively contribute to the development of safe, multimodal AI systems that augment human capabilities while minimizing potential risks.

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

## A. Core Contributions

We summarize our key contributions as follows:

- **Cognitively Inspired Evaluation Framework:** We design a four-stage benchmark consisting of Sensation, Perception, Understanding, and Reasoning, grounded in human cognition for a unified and interpretable evaluation of Omni-MLLMs.

- **Four-Level Intelligence Taxonomy:** We introduce a taxonomy that measures Task-, Modality-, Stage-, and Domain-Adaptive intelligence, enabling principled and fine-grained quantification of model capabilities.

- **Insightful Empirical Findings:** Evaluating 28 models, we report six key observations, including modality imbalances and stage-specific bottlenecks, highlighting current limitations and guiding future research directions.

## B. Discussion

### B.1. Existing Tasks

AVI-Bench builds upon many canonical audio-visual tasks, but these are used as foundational components rather than novel contributions. Our objective is not to create new tasks for their own sake; instead, we leverage well-studied tasks to systematically benchmark Omni-MLLMs' human-like audio-visual intelligence within a unified, cognitively grounded framework. At the Primitive Sensation level, we introduce the audio-visual sensation QA (AVSQA) task for the first time, incorporating with ASQA and VSQA to assess modality-adaptive capabilities.

### B.2. Use of Subsets

Using curated subsets of established tasks is a standard practice in benchmark design, as seen in General-Bench (ICML'25), SAVE-Bench (ICML'24), MMT-Bench (ICML'24), AVTrustBench (ICCV'25), AVHBench (ICLR'25), and OmnixR (ICLR'25). In AVI-Bench, selected subsets allow us to focus on evaluating adaptive reasoning capabilities beyond commonly used tasks in large model training, such as VQA and AQA. By carefully selecting and formatting these tasks, we ensure that they are informative for assessing human-like audio-visual intelligence rather than merely reflecting model exposure during pretraining. Additionally, over 62% of AVI-Bench samples are newly annotated, and evaluation reliability is improved through response masking (see Section D.2) and additional answer options such as "same" or "not sure."

### B.3. Data Scale

AVI-Bench is intentionally compact, designed to evaluate human-like cognition rather than train large models. With 5,864 carefully curated samples, it is larger than many prior benchmarks such as WorldQA (Zhang et al., 2024) (1,007), OmniBench (Li et al., 2025c) (1,142), OmniXR (Chen et al., 2024) (1,800), AVCaps (Sudarsanam et al., 2025) (2,061), WordSense (Hong et al., 2025) (3,172), AV-Odyssey (Gong et al., 2024) (4,555), and AVHBench (Sung-Bin et al., 2024) (5,302), yet remains small enough to focus on reasoning fidelity. Human performance on AVI-Bench averages 92.6, far surpassing Gemini-2.5-Pro (57.2), demonstrating that the benchmark is closely aligned with normal human capability while presenting a substantial challenge for current MLLMs (see Table 4). Statistical significance of these evaluations is further validated in Section E.2.

## C. Additional Backgrounds

### C.1. Audio-Visual Intelligence and Human Cognition

Audio-visual intelligence refers to the integrated ability to perceive, interpret, and reason about audio and visual inputs, which are two primary sensory channels accounting for more than 90% of human perception in the real world (Stein & Stanford, 2008). This ability supports essential cognitive functions such as scene understanding, event recognition, and cross-modal reasoning. From the perspective of human cognition, the human brain processes multimodal information through a hierarchical architecture. Early sensory processing, known as sensation, takes place in modality-specific regions. The auditory cortex, located in the temporal lobe (NCBI, a), encodes basic acoustic features such as pitch and volume. Meanwhile, the visual cortex in the occipital lobe (NCBI, b) extracts fundamental visual features including shape, edges, and color. These low-level signals are then transmitted to higher-order cortical areas where abstraction and integration occur. For example, the inferior temporal cortex supports category-level semantic recognition (Conway, 2018), and the parietal

lobe contributes to spatial localization and alignment across modalities (Macaluso et al., 2003). At the highest level, the prefrontal cortex coordinates semantic understanding, decision-making, and reasoning across sensory modalities, enabling abstract and goal-directed multimodal cognition (Friedman & Robbins, 2022).

## C.2. Audio-Visual Intelligence of Omni-MLLMs

As illustrated in Table 6, we provide a concise summary of the tasks included in AVI-Bench, grounded in human cognitive capabilities and their associated brain regions. However, in practical terms, current training methods for Omni-MLLMs mainly involve data with high-level semantic tasks such as multimodal captioning and question answering. As a result, these models tend to perform better on stages corresponding to perception, understanding, and reasoning. To better reflect this fact, AVI-Bench repositions the initial stage of multimodal processing, traditionally called sensation, to occur after reasoning. This adjustment enables a focused evaluation of the unfamiliar domain robustness of Omni-MLLMs, following extensive training on semantically rich tasks in common domains. In particular, it evaluates the models on challenges involving geometry, texture, volume, and clarity, which humans can handle effortlessly. This staged evaluation framework, inspired by human cognitive processes, offers a principled approach to assessing how modern multimodal AI exhibit human-like audio-visual intelligence.

*Table 6.* A bottom-up task taxonomy for AVI-Bench, rooted in human cognitive principles and structured according to the cortical processing hierarchy. It encompasses capabilities spanning from sensation to reasoning, covering multiple distinct dimensions such as **audio** versus **visual** sensation, **local** versus **global** perception, **acquisitive** versus **narrative** understanding, and **fine-grained** versus **coarse-grained** reasoning across both audio and visual modalities.

| Task ID | Task Name | Modality | Capability | Brain Lobe |
|---|---|---|---|---|
| **(Primitive) Sensation** | | | | |
| ASQA | Audio Sensation Question Answering | A | Audio Sensation | Temporal |
| VSQA | Visual Sensation Question Answering | I,V | Visual Sensation | Occipital |
| AVSQA | Audio-Visual Sensation Question Answering | A,I,V | Cross-modal Sensation | Temporal, Occipital |
| **Perception** | | | | |
| AMIC | Audio Multi-instance Classification | A | Audio Perception | Temporal |
| VMIC | Video Multi-instance Classification | I | Visual Perception | Temporal |
| AVL | Audio-Visual Localization | A,I | Cross-modal Local Perception | Parietal |
| AVM | Audio-Visual Matching | A,V | Cross-modal Global Perception | Parietal |
| **Understanding** | | | | |
| VAR | Visual-ref Audio Retrieval | A,I | Cross-modal Acquisitive Understanding | Frontal |
| AVR | Audio-ref Visual Retrieval | A,I | Cross-modal Acquisitive Understanding | Frontal |
| AVC | Audio-Visual Captioning | A,V | Cross-modal Narrative Understanding | Frontal |
| **Reasoning** | | | | |
| VAH | Visual-ref Audio Hallucination | A,V | Cross-modal Resistance Reasoning | Frontal Lobe |
| AVH | Audio-ref Visual Hallucination | A,V | Cross-modal Resistance Reasoning | Frontal Lobe |
| AVQA | Audio-Visual Question Answering | A,V | Cross-modal Coarse-grained Reasoning | Frontal Lobe |
| AVLG | Audio-Visual Language Grounding | A,V | Cross-modal Fine-grained Reasoning | Frontal Lobe |

# D. Additional Experiments

## D.1. Unimodal Ablation

As shown in Table 7, we conduct a unimodal ablation study across the AVL, AVC, AVH, VAH, AVQA, AVLG, and AVSQA tasks to assess the actual contribution of audio-visual modalities to task performance. Specifically, for the VAH task, we remove visual inputs and retain only audio. For all other tasks, we remove audio inputs and preserve only visual information. As expected, incorporating multimodal information leads to substantial performance improvements on the AVSQA task compared with unimodal inputs. However, for the AVH and VAH tasks, the additional modality aims to introduce multimodal inconsistencies. By removing the modality responsible for these inconsistencies, these tasks effectively simplify into single-modality VQA or AQA problems. This simplification results in performance gains for most models under unimodal conditions. These findings suggest that although Omni-MLLMs show promising results on AVH and VAH, their performance remains vulnerable to hallucinations caused by inconsistent or even conflicting multimodal signals.

Results on the AVL task reveal a more nuanced pattern. Robust models, including those in the Gemini series, Qwen series,

*Table 7.* Comparison of ***unimodal and multimodal*** performance on AVI-Bench. Only tasks that can be completed by unimodal models are included. The models are listed in alphabetical order. *Note:* to ensure a controlled *uni.*-vs-*mm.* comparison, AVC scores in this table use the original captioning metric, i.e., the mean of METEOR, ROUGE_L, CIDEr, and SBERT-similarity, applied identically to both columns. The AVC column in Table 3 uses the updated **FENSE** captioning evaluation; therefore the *mm.* AVC numbers here are not directly comparable to Table 3, while all other tasks share the same scoring as Table 3.

| Model | Params. | AVL | | AVC | | AVH | | VAH | | AVQA | | AVLG | | AVSQA | |
|---|---|---|---|---|---|---|---|---|---|---|---|---|---|---|---|
| | | uni. | mm. | uni. | mm. | uni. | mm. | uni. | mm. | uni. | mm. | uni. | mm. | uni. | mm. |
| Baichuan-Omni | 7B | 11.63 | 12.89 | 25.65 | 23.51 | 79.20 | 74.00 | 72.40 | 54.40 | 57.08 | 55.59 | 02.34 | 02.75 | 07.22 | 13.40 |
| Gemini-1.5-Flash | - | 14.14 | 36.33 | 25.11 | 24.89 | 86.00 | 84.80 | 68.80 | 66.40 | 68.55 | 69.80 | 31.35 | 30.29 | 02.06 | 17.01 |
| Gemini-1.5-Pro | - | 26.53 | 43.63 | 27.61 | 27.22 | 84.40 | 85.20 | 62.40 | 62.80 | 67.51 | 67.23 | 29.17 | 29.54 | 05.16 | 23.71 |
| Gemini-2.0-Flash | - | 13.25 | 37.93 | 28.36 | 27.75 | 83.60 | 84.80 | 75.60 | 75.20 | 68.13 | 68.51 | 29.99 | 27.61 | 06.70 | 26.80 |
| Gemini-2.5-Flash | - | 21.01 | 39.18 | 28.93 | 29.78 | 80.00 | 79.20 | 77.20 | 70.80 | 68.47 | 72.01 | 31.79 | 32.79 | 04.23 | 24.74 |
| Gemini-2.5-Pro | - | 27.50 | 39.13 | 29.03 | 27.79 | 84.00 | 86.40 | 79.20 | 80.80 | 71.38 | 73.93 | 40.57 | 35.08 | 06.70 | 16.50 |
| GPT-4o | - | 16.39 | 20.66 | 28.17 | 27.66 | 77.20 | 87.60 | 27.20 | 64.80 | 54.19 | 54.03 | 21.89 | 21.07 | 01.03 | 09.75 |
| GPT-4o-Mini | - | 13.86 | 18.49 | 24.36 | 25.39 | 79.20 | 74.80 | 08.00 | 34.80 | 49.69 | 50.39 | 17.13 | 17.47 | 05.16 | 06.19 |
| Human-Omni | 0.5B | 05.62 | 01.28 | 19.21 | 24.87 | 30.00 | 36.80 | 00.40 | 43.60 | 38.89 | 37.08 | 00.07 | 00.12 | 06.62 | 14.43 |
| Human-Omni | 7B | 05.86 | 01.96 | 25.80 | 33.79 | 86.40 | 86.00 | 71.60 | 57.20 | 52.63 | 56.54 | 00.03 | 00.02 | 06.19 | 15.46 |
| ImageBind-LLM | 7B | 05.21 | 01.42 | 22.19 | 23.11 | 50.40 | 40.40 | 47.60 | 50.40 | 28.09 | 31.11 | 00.08 | 00.02 | 00.00 | 00.52 |
| IXC2.5-OL | 7B | 07.07 | 05.68 | 29.22 | 29.52 | 78.00 | 80.40 | 07.20 | 58.40 | 55.09 | 53.23 | 13.24 | 13.27 | 00.00 | 01.03 |
| NExT-GPT | 7B | 05.58 | 01.28 | 20.65 | 12.90 | 31.60 | 11.60 | 51.60 | 29.60 | 29.36 | 16.75 | 01.39 | 01.47 | 05.16 | 07.73 |
| Ola | 7B | 16.76 | 17.88 | 25.34 | 30.06 | 67.60 | 79.20 | 81.20 | 83.20 | 60.97 | 58.21 | 06.77 | 07.07 | 00.52 | 11.86 |
| OneLLM | 7B | 06.54 | 02.30 | 28.23 | 28.02 | 85.20 | 79.60 | 62.40 | 60.80 | 39.23 | 40.59 | 00.00 | 00.00 | 08.25 | 19.07 |
| PandaGPT | 13B | 06.04 | 01.34 | 22.87 | 17.63 | 48.00 | 38.80 | 48.00 | 54.00 | 32.97 | 28.22 | 01.26 | 00.08 | 05.16 | 07.73 |
| PandaGPT | 7B | 05.73 | 01.46 | 23.57 | 16.24 | 51.60 | 43.20 | 56.80 | 56.00 | 35.79 | 35.12 | 00.16 | 00.08 | 08.25 | 07.73 |
| Phi-4-Multimodal | 5.6B | 07.98 | 10.61 | 32.14 | 32.49 | 88.00 | 86.80 | 58.00 | 60.40 | 56.04 | 53.75 | 02.48 | 02.27 | 00.00 | 00.52 |
| Qwen-Omni-Turbo | - | 11.76 | 19.21 | 32.14 | 36.06 | 76.58 | 80.40 | 72.16 | 74.40 | 63.52 | 63.71 | 08.80 | 08.86 | 13.92 | 21.65 |
| Qwen2.5-Omni | 7B | 12.80 | 19.36 | 32.16 | 35.98 | 79.43 | 82.40 | 75.77 | 77.60 | 63.71 | 64.29 | 08.69 | 08.74 | 13.92 | 21.13 |
| R1-Omni | 0.5B | 06.27 | 05.01 | 17.81 | 02.02 | 09.20 | 18.80 | 08.00 | 20.00 | 25.04 | 07.36 | 00.00 | 00.00 | 04.12 | 00.00 |
| Reka-Flash | 21B | 16.70 | 17.28 | 28.73 | 28.21 | 64.40 | 66.80 | 58.40 | 48.40 | 54.03 | 57.55 | 22.73 | 20.55 | 00.00 | 01.55 |
| UniMoE | 7Bx4 | 08.22 | 01.42 | 12.47 | 13.61 | 63.20 | 59.60 | 44.80 | 50.80 | 36.05 | 34.43 | 10.23 | 07.36 | 03.09 | 05.67 |
| Video-LLaMA2 | 7B | 06.61 | 02.68 | 28.44 | 30.43 | 32.40 | 44.80 | 86.80 | 70.00 | 64.43 | 44.16 | 02.35 | 00.86 | 15.46 | 21.65 |
| Video-salmonn | 13B | 05.94 | 02.04 | 30.99 | 33.35 | 75.20 | 57.60 | 61.60 | 58.00 | 42.03 | 42.73 | 00.71 | 01.38 | 07.73 | 06.70 |
| VITA-1.5 | 7B | 10.16 | 12.54 | 20.64 | 16.19 | 86.80 | 74.00 | 59.20 | 58.00 | 30.28 | 34.11 | 01.65 | 01.45 | 02.58 | 03.61 |
| X-Instruct-BLIP | 7B | 05.64 | 01.37 | 27.32 | 25.53 | 32.80 | 38.40 | 49.60 | 15.20 | 36.32 | 36.07 | 00.00 | 00.00 | 13.92 | 12.37 |

Ola, and Reka-Flash, benefit from multimodal inputs and outperform their unimodal counterparts. In contrast, models with weaker baseline performance on AVL, such as UniMoE, Video-LLaMA2, Human-Omni, and R1-Omni, achieve better results when relying solely on unimodal inputs. This contrast indicates that high-performing Omni-MLLMs can effectively leverage auditory cues to enhance visual grounding in this challenging setting, whereas less capable models suffer performance degradation when additional modalities are introduced. A concerning trend emerges from the AVLG results, where many models perform similarly or even better with unimodal inputs than with multimodal ones. Further analysis of the AVQA task reveals a comparable pattern, with unimodal inputs frequently yielding superior outcomes. These observations, especially in the reasoning-stage tasks, highlight significant limitations in current Omni-MLLMs to execute complex spatio-temporal reasoning with cross-modal synergy. To address these, future research should focus on improving the balanced audio-visual intelligence through enhanced spatio-temporal modeling.

### D.2. Response Masking with Double-confirmation

For the primitive sensation evaluation stage, we adopt a multiple-choice question (MCQ) format to enable efficient assessment. Due to the distinct characteristics of the unfamiliar domain data with low semantics, we introduce a double-confirmation mechanism with masked responses to enhance evaluation accuracy. Specifically, each question is accompanied by an additional confirmation query, such as "Can you see any object?" or "Can you hear any sound?" Both the original question and its confirmation include distractor options designed to verify whether the model truly understand the visual or audio input, rather than relying on hallucination or guessing. Finally, a response is considered correct only if the model answers both the original and the confirmation questions accurately.

As shown in Table 8, we present the results for the "Primitive Sensation" task under two conditions: using double-confirmation or not (i.e., using a one-time response). The one-time response approach generally yields higher performance scores. However, an intriguing and counterintuitive observation arises when comparing models such as Human-Omni-

0.5B and Human-Omni-7B, as well as PandaGPT-7B and PandaGPT-13B. In the absence of double-confirmation, these smaller-parameter models surprisingly outperform their larger counterparts on the ASQA task (PandaGPT-7B further leads PandaGPT-13B on the overall average). Notably, Human-Omni-0.5B, with only 0.5 billion parameters, achieves a score of 31.47% on the ASQA task, surpassing the strong closed-source model Gemini-1.5-Pro and is comparable to Gemini-2.0-Flash. By employing the double-confirmation mechanism, we aim to minimize hallucinations on unfamiliar domain data and to explore the true adaptation capabilities of Omni-MLLMs. Although this approach helps reduce erroneous assessments, completely eliminating such issues within the MCQ format remains fundamentally challenging. Future work should consider adopting more comprehensive and flexible question-answering formats.

*Table 8.* Comparison of the use of a ***double-confirmation mechanism***. "w/" indicates the use of double confirmation, while "w/o" indicates no double-confirmation applied. Models are ranked according to their average performance with double-confirmation.

| Models | Params. | ASQA | | VSQA | | AVSQA | | avg. | |
|---|---|---|---|---|---|---|---|---|---|
| | | w/ | w/o | w/ | w/o | w/ | w/o | w/ | w/o |
| Gemini-2.5-Pro | - | 29.48 | 39.44 | 62.67 | 63.83 | 16.50 | 18.56 | 36.22 | 40.61 |
| Gemini-2.5-Flash | - | 23.11 | 34.26 | 44.04 | 64.84 | 24.74 | 24.74 | 30.63 | 41.28 |
| Gemini-2.0-Flash | - | 21.51 | 31.47 | 40.13 | 59.80 | 26.80 | 26.80 | 29.48 | 39.36 |
| Qwen-Omni-Turbo | - | 21.51 | 36.25 | 33.40 | 55.34 | 21.65 | 21.65 | 25.52 | 37.75 |
| Qwen2.5-Omni | 7B | 21.12 | 35.46 | 31.51 | 55.34 | 21.13 | 21.13 | 24.59 | 37.31 |
| Gemini-1.5-Pro | - | 17.93 | 28.29 | 28.27 | 55.45 | 23.71 | 27.32 | 23.30 | 37.02 |
| Gemini-1.5-Flash | - | 21.91 | 34.66 | 24.06 | 55.40 | 17.01 | 19.59 | 21.00 | 36.55 |
| Baichuan-Omni | 7B | 24.70 | 39.04 | 23.56 | 43.89 | 13.40 | 20.10 | 20.56 | 34.34 |
| Video-LLaMA2 | 7B | 24.30 | 30.28 | 15.06 | 36.00 | 21.65 | 30.93 | 20.34 | 32.40 |
| OneLLM | 7B | 24.70 | 30.68 | 10.85 | 28.93 | 19.07 | 20.62 | 18.21 | 26.74 |
| Ola | 7B | 11.95 | 27.89 | 27.47 | 46.82 | 11.86 | 14.43 | 17.09 | 29.71 |
| Human-Omni | 7B | 15.54 | 28.29 | 19.47 | 40.08 | 15.46 | 40.72 | 16.82 | 36.36 |
| GPT-4o | - | 00.40 | 02.79 | 40.27 | 60.27 | 09.75 | 04.12 | 16.81 | 22.39 |
| Reka-Flash | 21B | 21.91 | 31.47 | 16.88 | 45.32 | 01.55 | 17.01 | 13.45 | 31.27 |
| UniMoE | 7Bx4 | 18.33 | 34.66 | 14.77 | 28.81 | 05.67 | 17.01 | 12.92 | 26.83 |
| PandaGPT | 13B | 13.15 | 21.12 | 15.55 | 27.14 | 07.73 | 13.92 | 12.14 | 20.72 |
| Vita-1.5 | 7B | 00.40 | 01.20 | 31.74 | 50.61 | 03.61 | 29.90 | 11.91 | 27.23 |
| Phi-4-Multimodal | 5.6B | 01.59 | 05.98 | 33.26 | 53.25 | 00.52 | 22.68 | 11.79 | 27.30 |
| NExT-GPT | 7B | 11.95 | 13.15 | 12.40 | 23.52 | 07.73 | 17.53 | 10.70 | 18.07 |
| GPT-4o-Mini | - | 00.00 | 01.20 | 25.85 | 44.07 | 06.19 | 11.86 | 10.68 | 19.04 |
| IXC2.5-OL | 7B | 04.38 | 15.14 | 25.43 | 45.27 | 01.03 | 18.04 | 10.28 | 26.15 |
| PandaGPT | 7B | 10.76 | 30.28 | 08.59 | 28.23 | 07.73 | 16.50 | 09.03 | 25.00 |
| Video-salmonn | 13B | 07.57 | 25.10 | 06.48 | 22.60 | 06.70 | 10.82 | 06.92 | 19.51 |
| Human-Omni | 0.5B | 00.00 | 31.47 | 04.56 | 20.57 | 14.43 | 21.13 | 06.33 | 24.39 |
| X-Instruct-BLIP | 7B | 00.00 | 15.14 | 06.14 | 26.77 | 12.37 | 13.40 | 06.17 | 18.44 |
| ImageBind-LLM | 7B | 03.98 | 22.71 | 01.29 | 23.14 | 00.52 | 17.01 | 01.93 | 20.95 |
| R1-Omni | 0.5B | 00.00 | 25.90 | 03.35 | 12.47 | 00.00 | 07.73 | 01.12 | 15.37 |

## D.3. Multi-stage Baselines

We designed eight multi-stage baselines by modularly combining three types of components. For the vision backbone, we employed either Qwen2-VL (denoted as v1) or its successor Qwen2.5-VL (v2). For the audio encoder, we considered Qwen-Audio (a1) and Qwen2-Audio (a2). Finally, for the language model, we adopted Qwen2.5 (q1) and Qwen3 (q2). These shorthand notations (v1/v2, a1/a2, q1/q2) are used throughout this section to concisely describe different model configurations. As shown in Table 9, our results highlight two important insights. First, upgrading the language backbone (e.g., from q1 to q2) consistently enhances both stage-adaptive and domain-adaptive performance, even when the accompanying audio or vision modules are relatively weak. This suggests that stronger language models provide a robust foundation for multimodal reasoning, mitigating limitations from less capable modalities. Second, the combination of more powerful audio and vision encoders with an advanced language decoder yields the most synergistic improvements, leading

*Table 9.* Performance of multi-stage baselines across four levels. These eight baseline configurations are reported under the original $\Delta_s$ formulation; the revised bottleneck-aware $\Delta_s$ of Sec. 5.3 is applied to the main 28-model cohort in Table 5, while the qualitative trends discussed below (audio bottleneck, language backbone effect) are preserved under either formulation.

| Model | L1 (Task-Adaptive) | L2 (Modality-Adaptive) | L3 (Stage-Adaptive) | L4 (Domain-Adaptive) |
|---|---|---|---|---|
| q1_a1_v1 | 22.52 | 21.27 | 16.16 | 8.58 |
| q1_a1_v2 | 22.35 | 20.87 | 14.35 | 8.73 |
| q1_a2_v1 | 30.81 | 26.08 | 23.12 | 13.74 |
| q1_a2_v2 | 29.87 | 25.28 | 21.05 | 13.15 |
| q2_a1_v1 | 23.75 | 23.31 | 19.01 | 13.66 |
| q2_a1_v2 | 23.88 | 23.16 | 19.63 | 15.41 |
| q2_a2_v1 | 30.52 | 30.50 | 23.57 | 14.26 |
| q2_a2_v2 | 30.92 | 29.87 | 24.56 | 16.42 |

to state-of-the-art performance across multiple evaluation stages. Together, these findings underscore the critical roles of language modeling capacity and cross-modal synergy in building effective multi-stage baselines.

### D.4. Multimodal Encoder Versus the LLM Backbone

To better understand the impact of modality enhancements on audio-visual intelligence, we systematically compare audio (a1 vs. a2) and visual (v1 vs. v2) encoders across different backbone configurations. Results in Table 9 show that audio remains the primary performance bottleneck. Replacing the weaker Qwen-Audio (a1) with Qwen2-Audio (a2) consistently improves performance across all four levels of intelligence. For example, under q1 and v2, L1 score increases from 22.35 to 29.87, and L4 score improves from 8.73 to 13.15. Similar gains are observed under q2 and v2 (L1: 23.88 $\rightarrow$ 30.92; L4: 15.41 $\rightarrow$ 16.42), highlighting the importance of stronger audio modeling, particularly when the language backbone is weaker. In contrast, upgrades to the visual encoder yield only modest improvements. When audio and language are fixed (e.g., q2_a2_v1 vs. q2_a2_v2), visual upgrades show limited gains, suggesting that visual enhancements contribute to unfamiliar-domain adaptation but exert a smaller overall impact.

Comparisons of language backbones also reveal three key findings: First, Qwen-3 consistently outperforms Qwen-2.5, particularly in stage- and domain-adaptive settings. Second, language strength is critical for reasoning and adaptation, even when paired with weaker modalities. Finally, with strong encoders (a2_v2), Qwen-3 demonstrates enhanced synergy and cognitive consistency.

## E. Additional Analysis

### E.1. Scores and Ranks

In this section, we provide supplementary results on task scores and ranks for each model across different evaluation stages. As shown in Figures 4 and 5, models generally perform better on vision-preferred tasks, particularly in the perception task VMIC, the reasoning task AVH, and the primitive sensation task VSQA. This observation supports our finding that the current development of visual and audio intelligence in Omni-MLLMs is imbalanced, with most models predominantly acting as visual specialists.

Furthermore, from Figure 4, it is evident that Omni-MLLMs struggle on tasks centered around grounding capabilities, such as AVL and AVLG. Notably, in the AVSQA task, the top-performing model Gemini-2.5-Pro achieves a score of only 16.5%, which is significantly lower than its performance on other sensation tasks and even lower than models with comparatively weaker overall capabilities (e.g., Gemini-2.5-Flash and Gemini-2.0-Flash). This indicates that Gemini-2.5-Pro lacks robustness in unfamiliar audiovisual tasks. Moreover, the highest score achieved by any evaluated model on AVSQA is only 26.8%, highlighting the difficulty Omni-MLLMs face in demonstrating strong audiovisual intelligence under unfamiliar-domain conditions.

### E.2. Statistical Significance

As shown in Table 10, we conduct the Shapiro-Wilk normality test on the performance scores of each evaluation stage as well as the overall performance to assess whether these data conform to the normality assumption. Specifically, we compute the

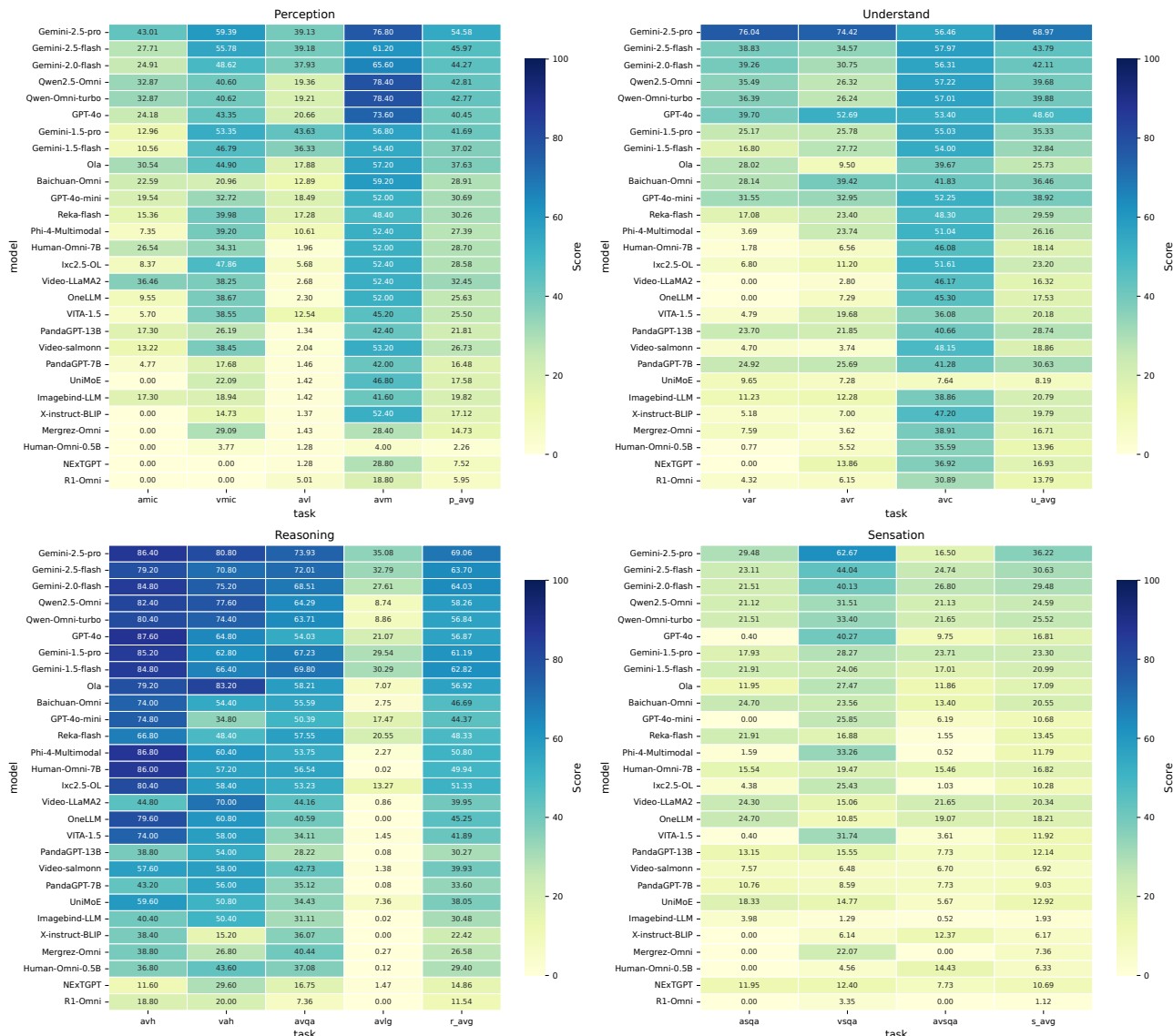

*Figure 4.* Task scores per model across different evaluation stages. Zoom-in for better visualization.

Shapiro-Wilk test statistic and corresponding p-values for the performance scores of the perception, understanding, reasoning, primitive sensation stages, and the average performance. The results indicate that, with the exception of Understanding ($p = 0.041$, marginally below the 0.05 threshold but well above the conservative 0.005 threshold), the p-values for all stages and the overall scores are substantially greater than the conventional significance thresholds, indicating no evidence to reject the null hypothesis of normality. Consequently, these performance scores can be considered approximately normally distributed. Based on this finding, it is appropriate to apply parametric statistical methods that assume normality for further correlation analyses and statistical inference, thereby ensuring the scientific rigor and accuracy of the results. Moreover, the approximate normality of the data suggests that the performance metrics are evenly distributed across samples without severe skewness or outliers, providing a robust statistical foundation for comprehensive evaluation of model performance.

In addition, as shown in Table 11, we perform Pearson correlation analysis to quantitatively assess the linear relationships between individual stage performances and the overall average performance. The motivation behind this analysis is to understand how each stage contributes to or aligns with the overall model capability. Using Pearson correlation coefficients, we compute the correlation strength (r) and statistical significance (p-value) between the scores of perception, understanding, reasoning, and primitive sensation stages and the overall average. The results show strong positive correlations (r values

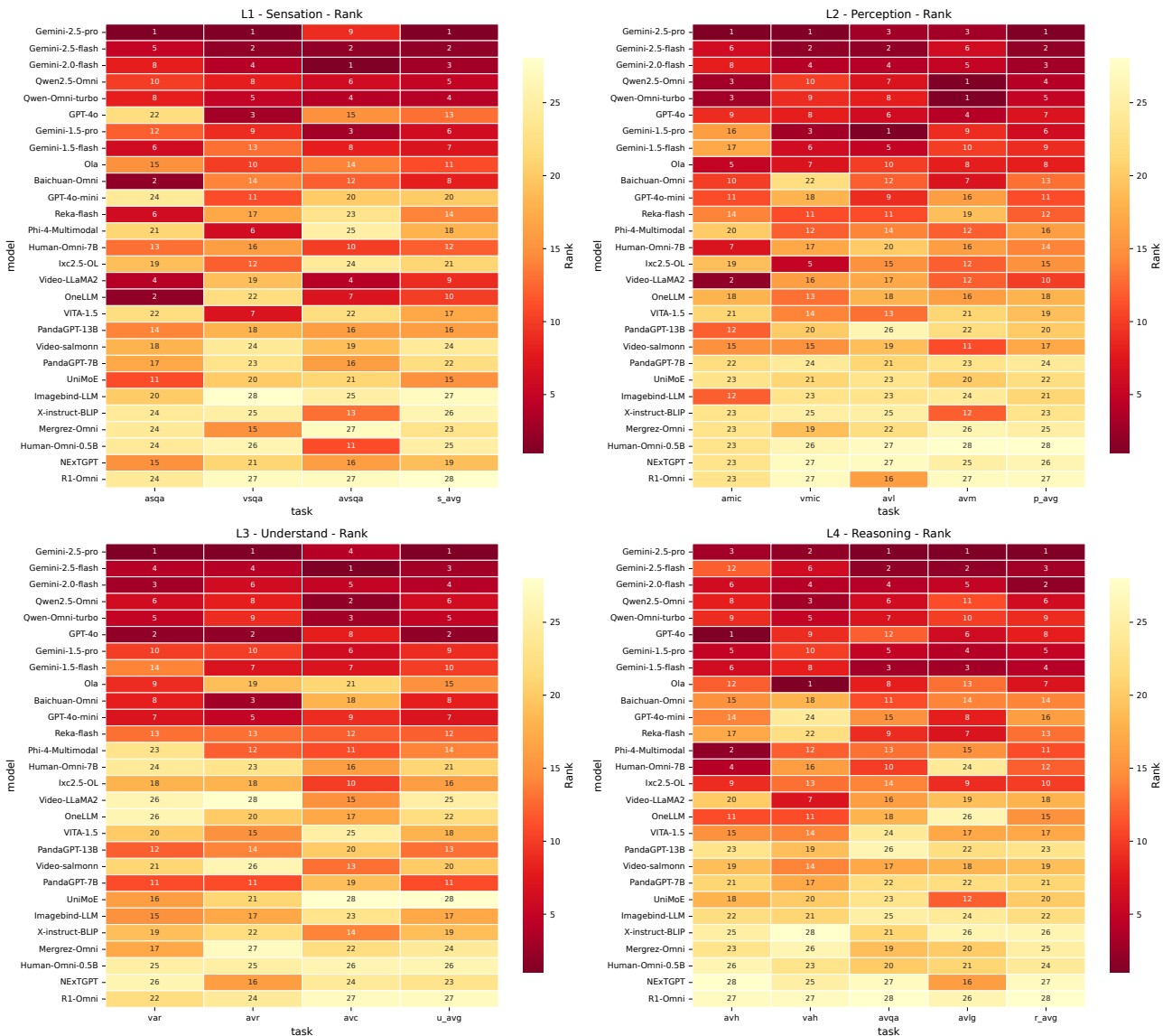

*Figure 5.* Task ranks per model across different evaluation stages. Zoom-in for better visualization.

ranging from 0.725 to 0.973) with highly significant p-values (all near zero), indicating that performance across individual stages is closely aligned with the overall model performance. These findings demonstrate the consistency and relevance of the model's capabilities across different stages.

### E.3. Stability and Reproducibility

All model evaluations in AVI-Bench were conducted using the same random seed (`seed=42`) and standardized prompts to ensure fair and consistent testing conditions. This approach, also adopted by *MMMU* (CVPR'24 Oral) and *General-Bench* (ICML'25 Oral), avoids the complexity and subjectivity of tuning prompts per model. To test the impact of random seed on performance differences, we compared two baseline configurations: **Baseline M1:** Qwen2.5 (text), Qwen2-VL (vision), Qwen-Audio (audio); **Baseline M2:** Qwen3 (text), Qwen2.5-VL (vision), Qwen2-Audio (audio).

Table 12 indicates that seed variation has only a limited effect on relative model rankings. For fairness and consistency, we therefore adopt a fixed seed, shared base prompt, and unified prompt format across all models. While our primary focus is on AVI, we view instruction-following as a core capability that should be evaluated without manual prompt tuning.

*Table 10.* Shapiro-Wilk normality test results for each stage.

| Stage | Statistic (W) | p-value |
|---|---|---|
| Perception | 0.9811 | 0.8765 |
| Understanding | 0.9231 | 0.0413 |
| Reasoning | 0.9656 | 0.4692 |
| Primitive Sensation | 0.9655 | 0.4675 |
| Overall Average | 0.9818 | 0.8916 |

*Table 11.* Pearson correlation coefficients between each evaluation stage, overall average, and primitive sensation. **r** and **p** represent correlation coefficient and p-value, respectively.

| Stage | Correlation with Overall Average (r, p) | Correlation with Primitive Sensation (r, p) |
|---|---|---|
| Perception | (0.973, 0.000) | (0.856, 0.000) |
| Understanding | (0.879, 0.000) | (0.725, 0.000) |
| Reasoning | (0.944, 0.000) | (0.820, 0.000) |
| Primitive Sensation | (0.901, 0.000) | — |

# F. Evaluation Details

## F.1. Models

As shown in Table 13, we present the versions of the closed-source models accessed via API calls. Note that OpenAI has not yet released an omni-model API that integrates both visual and audio modalities. Therefore, we adopt a cascaded approach: first, generating captions using the corresponding audio-preview version (e.g., GPT-4o-audio-preview and GPT-4o-mini-audio-preview), and then passing them to the corresponding vision-language models (e.g., GPT-4o and GPT-4o-mini).

## F.2. Model Outputs Formatting

Due to the inherent uncertainty in output formats and relatively limited instruction-following capabilities of LLMs, especially early-stage multimodal models, we propose to employ a pure language LLM to standardize model outputs for easier evaluation. Specifically, we employ GLM-4-Flash (GLM et al., 2024) as a text-formatting model. Given the original outputs together with task-specific prompts containing format instructions, it generates unified and structured outputs.

## F.3. Compute Resource

For each model evaluated without API calls, the evaluation is performed on a single 80GB NVIDIA A800 GPU.

# G. Metrics

## G.1. Task Metrics

### G.1.1. AMIC

Given a dataset with $N$ samples, for the $i$-th sample, the model predicts a set of category-instance count pairs:

$$\hat{\mathbf{y}}_i = \{(\hat{c}_1, \hat{n}_1), (\hat{c}_2, \hat{n}_2), \dots\}, \tag{8}$$

where $\hat{c}_j$ denotes the predicted category and $\hat{n}_j$ its corresponding instance count. The ground truth label is:

$$\mathbf{y}_i = \{(c_1, n_1), (c_2, n_2), \dots\}. \tag{9}$$

*Table 12.* Impact of seed variation on baseline model performance across four evaluation levels. As with Table 9, scores are reported under the original $\Delta_s$ formulation; seed-induced rank-stability findings are independent of this choice.

| Model | L1 (Task-Adaptive) | L2 (Modality-Adaptive) | L3 (Stage-Adaptive) | L4 (Domain-Adaptive) |
|---|---|---|---|---|
| M1 (s=42) | 22.52 | 21.27 | 16.16 | 8.58 |
| M2 (s=42) | 30.92 | 29.87 | 24.56 | 16.42 |
| M1 (s=1024) | 22.09 | 21.30 | 16.21 | 8.64 |
| M2 (s=1024) | 30.81 | 28.09 | 25.22 | 16.53 |

*Table 13.* Model version codes for API calls.

| Omni-MLLMs | Version Code |
|---|---|
| Gemini-1.5-Flash | gemini-1.5-flash |
| Gemini-1.5-Pro | gemini-1.5-pro |
| Gemini-2.0-Flash | gemini-2.0-flash |
| Gemini-2.5-Flash | gemini-2.5-flash-preview-04-17 |
| Gemini-2.5-Pro | gemini-2.5-pro-preview-05-06 |
| GPT-4o | gpt-4o-2024-08-06 |
| GPT-4o-audio-preview | gpt-4o-audio-preview-2024-12-17 |
| GPT-4o-mini | gpt-4o-mini-2024-07-18 |
| GPT-4o-mini-audio-preview | gpt-4o-mini-audio-preview-2024-12-17 |
| Qwen-Omni-Turbo | qwen-omni-turbo-2025-03-26 |

**Semantic Matching Score:**  Define the predicted category set $\hat{C}_i = \{\hat{c}_j\}$ and the ground truth category set $C_i = \{c_j\}$. Let $\mathcal{C}_i = \hat{C}_i \cup C_i$, then for each category $c \in \mathcal{C}_i$, define binary label vectors:

$$y_c^{(i)} = \begin{cases} 1, & c \in C_i, \\ 0, & \text{otherwise}, \end{cases} \qquad \hat{y}_c^{(i)} = \begin{cases} 1, & c \in \hat{C}_i, \\ 0, & \text{otherwise}. \end{cases} \tag{10}$$

The semantic matching score for sample $i$ is defined as the F1 score between these vectors:

$$S_{\text{semantic}}^{(i)} = \text{F1}(\mathbf{y}^{(i)}, \hat{\mathbf{y}}^{(i)}), \tag{11}$$

then we get the overall semantic score based on the average over all samples:

$$S_{\text{semantic}} = \frac{1}{N} \sum_{i=1}^{N} S_{\text{semantic}}^{(i)}. \tag{12}$$

**Counting Error:**  For each sample $i$ and category $c \in C_i$, let the ground truth count be $n_c^{(i)}$ and predicted count be $\hat{n}_c^{(i)}$ if $c \in \hat{C}_i$, otherwise undefined.

Define the absolute counting error as:

$$e_c^{(i)} = \begin{cases} \left| \hat{n}_c^{(i)} - n_c^{(i)} \right|, & c \in \hat{C}_i, \\ \tau, & c \notin \hat{C}_i, \end{cases} \tag{13}$$

where $\tau$ is a predefined penalty constant for missing semantic predictions.

The average counting error per sample is:

$$E^{(i)} = \frac{1}{|C_i|} \sum_{c \in C_i} e_c^{(i)}. \tag{14}$$

The mean squared error (MSE) across the dataset is:

$$\text{MSE} = \frac{1}{N} \sum_{i=1}^{N} (E^{(i)})^2, \tag{15}$$

and the root mean squared error (RMSE) is:

$$\text{RMSE} = \sqrt{\text{MSE}}. \tag{16}$$

**Counting Score:** The counting score is computed by applying a nonlinear transformation to the root mean squared error (RMSE) between predicted and true instance counts. Specifically, it is defined as:

$$S_{\text{counting}} = 1 - \tanh(k \cdot \text{RMSE}), \tag{17}$$

where $\text{RMSE} \geq 0$ is the root mean squared error, and $k > 0$ is a scaling hyperparameter controlling the sensitivity of the score to the error magnitude.

Since RMSE is non-negative, the argument of the hyperbolic tangent function is always non-negative, resulting in: $\tanh(k \cdot \text{RMSE}) \in [0, 1)$, and consequently $S_{\text{counting}} \in (0, 1]$. The counting score is thus a monotonically decreasing function of RMSE, approaching 1 as RMSE approaches zero. If RMSE is invalid or negative, the counting score is set to zero.

**Final AMIC Score:** If the semantic score is zero, the counting score is set to zero, then the final AMIC score is the average of semantic and counting scores:

$$S = \frac{S_{\text{semantic}} + S_{\text{counting}}}{2}. \tag{18}$$

### G.1.2. VMIC

VMIC evaluates multi-instance visual classification and counting, where the input is an image or video frame and the model predicts categories present along with their instance counts. The semantic matching and counting error definitions follow those of AMIC, with the following key differences.

Firstly, the semantic category set used for evaluation is restricted to the ground-truth categories only:

$$\mathcal{C}_i = C_i, \tag{19}$$

reflecting the conservative nature of category annotations in the VMIC task. Due to the rich and complex visual content, the annotation process selectively includes only the more salient categories and their instances as required recognition targets. This approach ensures that the predicted categories and instance counts in both AMIC and VMIC remain on comparable scales, despite the typically strong visual capabilities of multimodal models. Secondly, VMIC uses recall instead of F1-score as the semantic classification metric. For the $i$-th sample, the semantic score is:

$$S_{\text{semantic}}^{(i)} = \text{Recall}(\mathbf{y}^{(i)}, \hat{\mathbf{y}}^{(i)}), \tag{20}$$

where $\mathbf{y}^{(i)}$ and $\hat{\mathbf{y}}^{(i)}$ are binary label vectors over $\mathcal{C}_i$ as defined in Equation 8 and 9.

### G.1.3. AVL

Given a dataset with $N$ samples, each sample consists of an audio input and a corresponding image. The task is to localize sound-emitting object in the image given an audio reference.

**Bounding Box Format and Preprocessing:** Each ground truth bounding box is represented as $\mathbf{g} = [x, y, w, h]$, where $(x, y)$ is the top-left corner and $(w, h)$ the width and height. These are converted to corner coordinates $[x_1, y_1, x_2, y_2]$ by:

$$\mathbf{g}' = [x, y, x + w, y + h]. \tag{21}$$

Predicted bounding boxes are normalized to $[0, 1]$, representing relative positions within the image width and height. To obtain absolute pixel coordinates, the normalized predictions are scaled by the original image dimensions $W$ and $H$:

$$\mathbf{p} = [x_1 W, y_1 H, x_2 W, y_2 H]. \tag{22}$$

**Matching and Intersection-over-Union Computation:**   For each ground truth bounding box $\mathbf{g}'_j$ in sample $i$, the algorithm searches among the unmatched predicted boxes of the same semantic category to find the one with the highest Intersection-over-Union (IoU):

$$\mathbf{p}_j = \arg \max_{\mathbf{p} \in \mathcal{U}_i^{(j)}} \text{IoU}(\mathbf{g}'_j, \mathbf{p}), \tag{23}$$

where

$$\text{IoU}(\mathbf{g}'_j, \mathbf{p}) = \frac{|\mathbf{g}'_j \cap \mathbf{p}|}{|\mathbf{g}'_j \cup \mathbf{p}|}. \tag{24}$$

If the maximum IoU is zero, the ground truth box $\mathbf{g}'_j$ is considered unmatched. Once matched, $\mathbf{p}_j$ is removed from the unmatched set $\mathcal{U}_i^{(j)}$. The per-sample mean IoU is computed by averaging IoU over *all* ground truth boxes by assigning zero IoU to unmatched instances:

$$\text{mIoU}_i = \frac{1}{|G_i|} \sum_{j=1}^{|G_i|} \text{IoU}(\mathbf{g}'_j, \mathbf{p}_j), \tag{25}$$

where $|G_i|$ is the total number of ground truth instances in sample $i$.

**Instance Error:**   The instance error for sample $i$ is the number of unmatched ground truth boxes:

$$E^{(i)} = |G_i| - M_i, \tag{26}$$

where $M_i$ is the number of matched ground truth instances.

The dataset-level root mean squared instance error (RMSE) is:

$$\text{RMSE} = \sqrt{\frac{1}{N} \sum_{i=1}^{N} (E^{(i)})^2}. \tag{27}$$

**Instance Score:**   The RMSE is transformed into an instance score $S$ (i.e., the counting score) according to Equation 17.

**Final Score:**   The overall final score combines the average mean IoU and the instance score with a weighting factor $\alpha$ set to 0.7 by default:

$$S_{\text{final}} = \alpha \cdot \frac{1}{N} \sum_{i=1}^{N} \text{mIoU}_i + (1 - \alpha) \cdot S. \tag{28}$$

G.1.4. VAR AND AVR

Consider a dataset with $N$ samples for the VAR task. For each sample $i$, the model predicts a set $\hat{R}_i$ of $m$ images from $n$ candidates based on the given audio, and the ground truth relevant set is $R_i$. In the case of the AVR task, we utilize a similar evaluation framework, but the roles of the modalities are reversed. Here, the audio serves as the input, while the related images are retrieved based on the audio reference.

**Average F1 Score:**   The average F1 score over the dataset is computed directly as:

$$\overline{F1} = \frac{1}{N} \sum_{i=1}^{N} \frac{2 \cdot |\hat{R}_i \cap R_i|}{|\hat{R}_i| + |R_i|}. \tag{29}$$

**Recall@k Metrics:**   Recall@$k$ measures the ability of the model to retrieve at least one relevant item within its top-$k$ predictions. For each sample $i$, we consider the top $k$ predicted images $\hat{R}_i^{[1:k]}$ and check whether there exists an intersection with the ground truth relevant set $R_i$. Formally, the per-sample Recall@$k$ is defined as

$$\text{Recall@}k_i = \begin{cases} 1, & \text{if } \hat{R}_i^{[1:k]} \cap R_i \neq \emptyset, \\ 0, & \text{otherwise.} \end{cases} \tag{30}$$

This indicator equals 1 if at least one correct image is retrieved in the top $k$ results, and 0 otherwise. The overall Recall@$k$ metric is computed as the average over all samples:

$$\overline{\text{Recall@}k} = \frac{1}{N} \sum_{i=1}^{N} \text{Recall@}k_i. \tag{31}$$

**Penalization and Confidence Adjustment:** In practical settings, retrieval models may produce outputs that contain a high degree of repetition or overly simplistic predictions, such as repeatedly returning the same images across different queries or merely counting without considering semantic content. These behaviors indicate a lack of meaningful understanding of the input data and degrade the quality of the retrieval. To mitigate such issues, we introduce a penalization mechanism that reduces the evaluation scores of models generating repetitive or non-diverse outputs:

$$\text{r} = \frac{N - |\{\hat{R}_i\}|}{N}, \tag{32}$$

where $|\{\hat{R}_i\}|$ denotes the number of unique predicted sets. To avoid overly penalizing minor repetition, a threshold $\tau$ (default as 0.8) is imposed to clip the repeat rate:

$$\text{r} = \max(\tau, \text{r}), \tag{33}$$

then we can compute the penalty P applied to final scores as:

$$\text{P} = 1 - (\text{r} - \tau), \tag{34}$$

which yields a value in $(0, 1]$. As the repeat rate exceeds $\tau$, the penalty decreases, thereby lowering final metric scores for repetitive outputs.

Additionally, in our evaluation framework, we incorporate a confidence factor that reflects the reliability of each model prediction based on the size of the predicted set. This design addresses a common issue observed in practice: some models output an excessively large number of candidate indices, often including many irrelevant ones, which dishonestly inflates performance metrics. Specifically, we define if the predicted set size $|\hat{R}_i|$ exceeds a predefined threshold $b$ (set as 6 by default), the confidence assigned to that prediction is reduced to $d$ (with 0.3 as the default); otherwise, it is set to 1. Formally, the per-sample confidence $c_i$ is defined as:

$$c_i = \begin{cases} d, & \text{if } |\hat{R}_i| > b, \\ 1, & \text{otherwise.} \end{cases} \tag{35}$$

The overall confidence score C used for metric scaling is the average over all samples:

$$\text{C} = \frac{1}{N} \sum_{i=1}^{N} c_i. \tag{36}$$

By combining the repeat penalty and confidence factor, the evaluation better reflects both the diversity and reliability of model predictions, encouraging models to produce semantically meaningful and diverse retrieval outputs.

**Final Metrics** The final evaluation metrics for Recall@1, Recall@3, and F1 are scaled by the confidence and repeat penalty terms:

$$\text{Recall@1} = \overline{\text{Recall@1}} \times \text{C} \times \text{P}, \tag{37}$$

$$\text{Recall@3} = \overline{\text{Recall@3}} \times \text{C} \times \text{P}, \tag{38}$$

$$F1 = \overline{F1} \times \text{C} \times \text{P}, \tag{39}$$

and the overall average score is computed as:

$$\text{S} = \frac{\left(\frac{\text{Recall@1} + \text{Recall@3}}{2}\right) + F1}{2}. \tag{40}$$

### G.1.5. AVC

To evaluate captions produced for the audio-visual captioning task, we adopt the Fluency-ENhanced Sentence-bert Evaluation (FENSE) metric, which couples a semantic-similarity term with a learned fluency penalty so that errors such as incomplete sentences or repetitions are explicitly down-weighted. Consider a dataset of $M$ samples. For each sample $i \in \{1, \ldots, M\}$, the model emits a single caption $\hat{y}_i$ and the ground truth provides a set of $N_i$ reference captions $\mathcal{Y}_i = \{y_i^{(1)}, \ldots, y_i^{(N_i)}\}$. The semantic component is the maximum SBERT cosine similarity between $\hat{y}_i$ and the reference set:

$$s_i^{\text{sem}} = \max_{j \in \{1, \ldots, N_i\}} \text{SBERT\_Sim}(\hat{y}_i, y_i^{(j)}). \tag{41}$$

A pretrained fluency-error detector $D(\hat{y}_i) \in [0, 1]$ estimates the probability that $\hat{y}_i$ contains a fluency error. The per-sample FENSE score combines the two terms through a multiplicative penalty:

$$\text{FENSE}(\hat{y}_i, \mathcal{Y}_i) = \big(1 - \alpha \cdot D(\hat{y}_i)\big) \cdot s_i^{\text{sem}}, \tag{42}$$

with the standard penalty weight $\alpha = 0.9$. The reported AVC score is the corpus mean:

$$\text{Caption Score} = \frac{1}{M} \sum_{i=1}^{M} \text{FENSE}(\hat{y}_i, \mathcal{Y}_i). \tag{43}$$

### G.1.6. AVLG

In the AVLG task, given a video $V$ with $T$ frames, accompanying audio, and a referring expression, the goal is to localize the referenced object in each frame by predicting a bounding box. The ground truth (GT) provides, for each frame $t \in \{1, \ldots, T\}$, a bounding box $\mathbf{g}_t$. The model outputs, for each frame, a predicted bounding box $\hat{\mathbf{p}}_t$ normalized to the frame's width and height.

**Bounding Box Format and Preprocessing:** Ground truth bounding boxes are represented as:

$$\mathbf{g}_t = [x_t, y_t, w_t, h_t], \tag{44}$$

which are converted to:

$$\mathbf{g}'_t = [x_t, y_t, x_t + w_t, y_t + h_t], \tag{45}$$

denoting the $(x_1, y_1, x_2, y_2)$ format.

Predicted bounding boxes are asked to be normalized by evaluated models as:

$$\hat{\mathbf{p}}_t = [\hat{x}_{1,t}, \hat{y}_{1,t}, \hat{x}_{2,t}, \hat{y}_{2,t}] \in [0, 1]^4, \tag{46}$$

meaning they represent the relative position and scale within the frame, with values ranging between 0 and 1. Then during the evaluation, they are scaled back to original frame dimensions $W, H$ like we mentioned in Equation 22:

$$\hat{\mathbf{p}}'_t = [\hat{x}_{1,t}W, \hat{y}_{1,t}H, \hat{x}_{2,t}W, \hat{y}_{2,t}H]. \tag{47}$$

**Per-Frame Intersection-over-Union (IoU):** For each frame $t$, the Intersection-over-Union between ground truth and predicted bounding boxes is computed as

$$\text{IoU}_t = \frac{|\mathbf{g}'_t \cap \hat{\mathbf{p}}'_t|}{|\mathbf{g}'_t \cup \hat{\mathbf{p}}'_t|}. \tag{48}$$

If either the GT or prediction is missing for frame $t$, the IoU is defined as

$$\text{IoU}_t = \begin{cases} 1, & \text{if both GT and prediction are none,} \\ 0, & \text{otherwise.} \end{cases} \tag{49}$$

**Video-Level IoU Score:** The video-level IoU score is the average IoU over all frames:

$$\text{IoU}_{\text{video}} = \frac{1}{T} \sum_{t=1}^{T} \text{IoU}_t, \tag{50}$$

then given a dataset with $N$ videos, the final evaluation score is the mean video-level IoU:

$$\text{IoU}_{\text{final}} = \frac{1}{N} \sum_{i=1}^{N} \text{IoU}_{\text{video}}^{(i)}. \tag{51}$$

### G.1.7. AVM, AVH, VAH, AVQA

Given a dataset containing $N$ samples, each sample $i$ is associated with a ground-truth label $y_i$ and a predicted answer $\hat{y}_i$. The prediction $\hat{y}_i$ is considered correct if and only if it exactly matches the ground-truth label $y_i$, i.e.,

$$a_i = \begin{cases} 1, & \text{if } \hat{y}_i = y_i, \\ 0, & \text{otherwise.} \end{cases} \tag{52}$$

The overall accuracy is computed as the average of the per-sample correctness indicators:

$$\text{Accuracy} = \frac{1}{N} \sum_{i=1}^{N} a_i. \tag{53}$$

### G.2. Stage Metrics

As presented in Section 5, the main paper introduces our approach for evaluating models' human-like audio-visual intelligence. In this section, we provide a more detailed discussion of the rationale and motivation underlying the design of these metrics.

#### G.2.1. LEVEL-2: MODALITY-ADAPTIVE

**Definition:** Evaluates the balance between the model's performance on audio and visual tasks, encouraging the development of synergistic "audio-visual" understanding rather than modality-specific specialization.

**Motivation:** Instead of using a direct penalty term like $|A - V|$, we adopt the metric

$$\Delta_m = \frac{2|A - V|}{A + V} \tag{54}$$

based on two key considerations:

1. **Consistent scalar range:** This allows the metric to be combined with other levels while controlling for penalty magnitude.

2. **Relative difference:** $|A - V|$ only reflects absolute deviation, even under normalization. In contrast, $\Delta_m$ captures the *relative imbalance* between modalities, offering a more meaningful indicator of coordination.

We illustrate this with the following comparison:

Given the computation for Level-2:

$$S_M = (1 - p) \cdot S_T, \quad p = \Delta_m \tag{55}$$

which can be interpreted as a penalty coefficient that reduces the Level-1 score $S_T$ based on modality imbalance. As shown in Table 14 and Figure 6, even with the same $|A - V| = 0.2$, Model C and D receive very different penalties:

*Table 14.* Comparison of absolute and relative modality imbalance metrics. "Balanced" and "Unbalanced" refer to the performance gap between audio and vision, while "High" and "Low" represents the model overall performance.

| Model | A | V | $|A - V|$ | $\frac{2|A-V|}{A+V}$ |
|---|---|---|---|---|
| Model A (Balanced, High) | 0.9 | 0.9 | 0.0 | 0.000 |
| Model B (Balanced, Low) | 0.2 | 0.2 | 0.0 | 0.000 |
| Model C (Unbalanced, High) | 0.9 | 0.7 | 0.2 | 0.250 |
| Model D (Unbalanced, Low) | 0.3 | 0.1 | 0.2 | 1.000 |
| Model E (Unbalanced, Low, reversed D) | 0.1 | 0.3 | 0.2 | 1.000 |

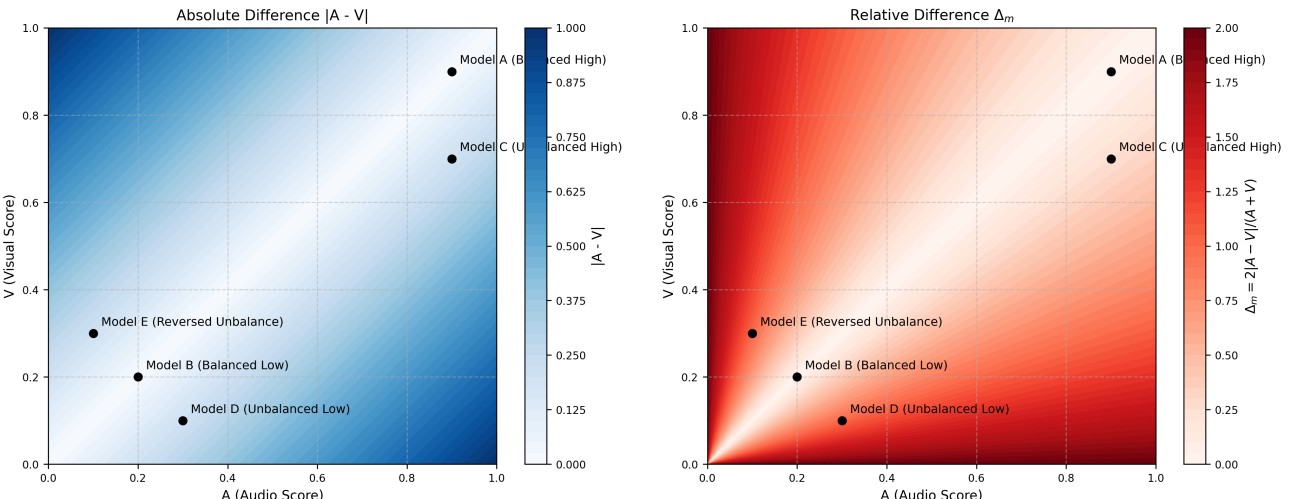

*Figure 6.* Visualized comparison of absolute and relative modality imbalance metrics among example data points.

- For Model C, the difference (0.2) represents $\sim 22\%$ for Audio (0.9) and $\sim 29\%$ for Vision (0.7), which is relatively moderate.

- For Model D, the difference (0.2) represents $\sim 66\%$ of Audio (0.3) and $\sim 200\%$ of Vision (0.1), which indicates a substantially greater imbalance.

This shows that using $\Delta_m$ provides a more nuanced evaluation of cross-modal coordination than $|A - V|$ alone. Moreover, we choose $A + V$ as the normalization base instead of using $A$, $V$, or $\max(A, V)$, because we view audio and visual performance as forming a *collaborative system*, not a competitive one.

### G.2.2. LEVEL-3: STAGE-ADAPTIVE

**Definition:** Measures whether reasoning is properly grounded in its perceptual and conceptual prerequisites, as defined by Eqs. (4)–(6) in Section 5.3.

**Motivation:** This design is motivated by Observation 2, which highlights the bottleneck effect of perception and understanding on reasoning. The bottleneck discrepancy $\Delta_s$ in Eq. (5) is asymmetric and directional: it penalizes only when reasoning runs ahead of its weakest cognitive prerequisite, reflecting the hypothesis that strong reasoning ought to be grounded in commensurately strong perception and understanding.

**Headroom normalization.** Different tasks in AVI-Bench have markedly different intrinsic difficulty; e.g., AVH and VAH are 3-option yes/no/not-sure decisions, whereas AVL is evaluated by mIoU over open bounding-box prediction. Without correction, this difficulty asymmetry would inflate a stage simply because its constituent tasks have higher random-guess floors. We therefore subtract a task-specific chance baseline $c_t$ and rescale the remaining headroom to $[0, 100]$ via Eq. (4).

*Table 15.* Task-specific chance baselines $c_t$ used in headroom normalization (Eq. (4)). Values are determined by task design (open-ended generation and mIoU $\rightarrow 0$; $k$-option MCQ $\rightarrow 100/k$) or by empirical measurement.

| Task | $c_t$ | Derivation |
|---|---|---|
| AMIC, VMIC, AVL, AVLG | 0.00 | open-ended generation / mIoU |
| AVM, AVH, VAH | 33.33 | 3-option {yes, no, not sure}, 1/3 |
| VAR, AVR | 10.00 | retrieval over 10 candidates, conservative noise floor |
| AVC | 13.06 | empirical FENSE on shuffled references |
| AVQA | 21.35 | sample-weighted (232 binary, 155 numeric, 82 cls) |
| ASQA | 29.66 | sample-weighted (342 4-opt, 120 3-opt, 40 counting) |
| VSQA | 26.70 | sample-weighted over VSQA-I + VSQA-V subtasks |
| AVSQA | 24.63 | sample-weighted (160 6-opt, 34 8-opt, 194 3-opt) |

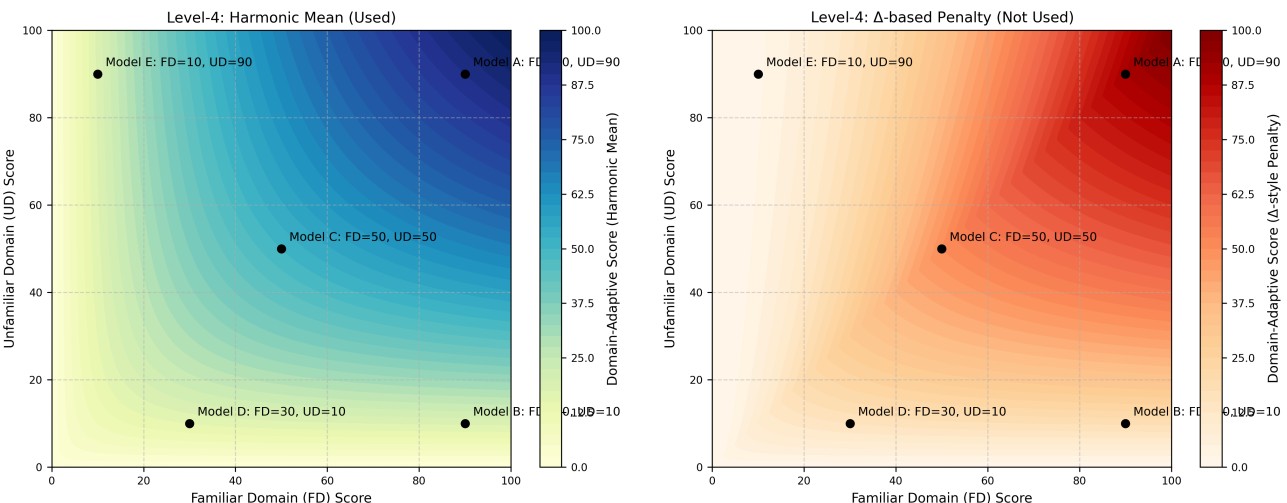

*Figure 7.* Visualized comparison of using harmonic mean and $\Delta$-based penalty to calculate Level-4 score.

Crucially, $c_t$ is determined by the structure of task $t$ itself, not by the cohort of evaluated models, so adding new models in future work does not retroactively change reported scores.

### G.2.3. LEVEL-4: DOMAIN-ADAPTIVE

**Definition:** Assesses the model's ability to adapt its familiar-domain capabilities to unfamiliar-domain scenarios, reflecting its potential for human-like cross-domain robustness.

**Motivation:** Level-4 focuses on robustness in unfamiliar-domains compared to mainstream training distributions. To reflect this lower-bound performance emphasis, we use the harmonic mean (see Eq. (7)), unlike Level-2 and Level-3 which prioritize balance via relative difference.

As shown in Figure 7, we can compare Model B and Model E. When using a $\Delta$-based penalty to compute the Level-4 score (right plot), the results become asymmetric between the two models. In contrast, the use of the harmonic mean ensures that the overall score is determined by the weaker component, reinforcing the principle that a model should perform well in both familiar and unfamiliar domains.

## H. Pipeline and Quality Control

As illustrated in Figure 8, we conduct the selection of tasks and capabilities to evaluate based on human cognition. For tasks in the perception-understanding-reasoning chain, we collect publicly available, easily accessible, and modifiable datasets from the web. These datasets are then restructured or re-annotated according to the task definitions specified in AVI-Bench. For the primitive sensation evaluation, which involves unfamiliar domain data which is far different from the commonly

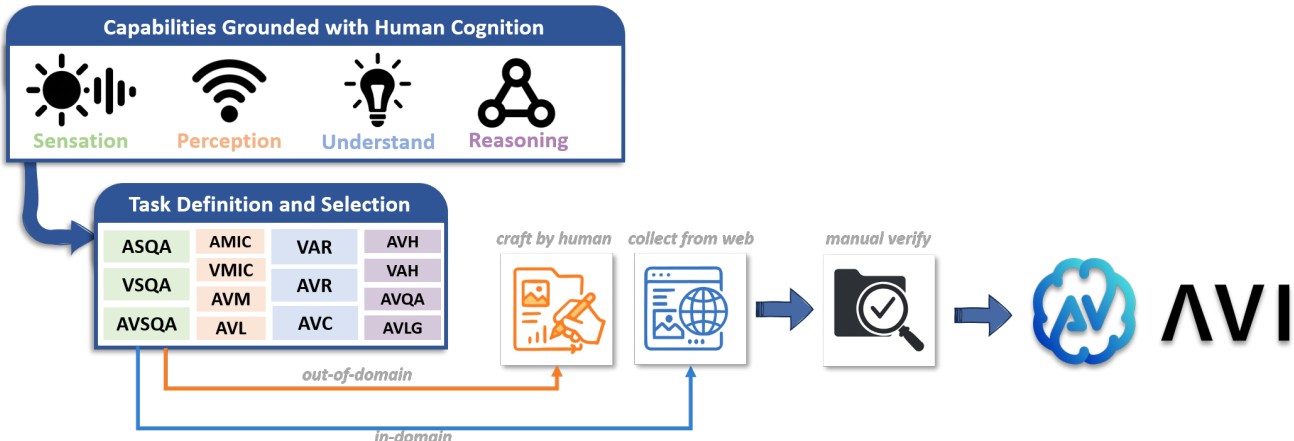

*Figure 8.* AVI-Bench construction pipeline. The media data collected online is assigned as familiar domain data with high semantics, while the manually constructed media data is considered unfamiliar domain data with low semantics. Both types will undergo manual verification, and for the online collected data, re-annotation and organization will be required as necessary.

used training domain, we construct the dataset entirely from scratch to ensure that it remains uncontaminated and unseen by any model.

To guarantee the data quality, regardless of whether the data are externally sourced (including both re-annotating and preserving original annotations) or entirely self-constructed, every dataset undergoes meticulous manual verification. We carefully screen all samples to remove erroneous or anomalous data present in the original sources. This rigorous quality control ensures that only high-quality and reliable data are used for evaluation, thereby guaranteeing the accuracy and fairness of the benchmark results.

In detail, we report the data source and quality control for both the familiar and unfamiliar domain data:

### H.1. Familiar-Domain Data

These datasets are sourced from the web and cover perception, understanding, and reasoning tasks. It reflects high-level semantics typical in model training. Some subsets are repurposed from leading benchmarks:

- AVL: AVS-Bench (ECCV'22) (Zhou et al., 2022), AVISeg (CVPR'25) (Guo et al., 2025)

- AVLG: Ref-AVS (ECCV'24) (Wang et al., 2024d)

- AVM, AVH, VAH: AVHBench (ICLR'25) (Sung-Bin et al., 2024)

- AVC: AVCaps (Sudarsanam et al., 2025), AVHBench

- AVQA: Music-AVQA (CVPR'22) (li et al., 2022)

These datasets are widely adopted for training and evaluation in the audio-visual domain. All data are standardized into the following JSON format:

```
{
    "id": 0000,
    "task": "AMIC",
    "input": {
        "question": {
            "prompt": "...",
            "text": "...",
            "options": "..."
        },
```

```
        "video": "...",
        "image_list": ["..."],
        "audio_list": ["..."]
    },
    "output": {
        "question_answer": "...",
        "pred_bbox": "..."
    }
}
```

After standardization, Group-A manually reviewed all familiar domain data, verifying both auditory and visual elements to ensure accuracy through a rigorous double-checking process.

### H.2. Unfamiliar-Domain Data (PriSe)

These datasets were constructed fully offline without reusing real-world content, focusing on low-semantic and rare distributions.

- Group-B constructed all PriSe data offline, including custom-generated images, synthesized audio, and composed videos, without using any real-world content.

- Before annotation, Group-C provided Group-B with example data for each task to ensure consistency.

- **First review:** Upon completion of 10% of the data, Group-C reviewed all samples, summarized issues, and provided feedback to Group-B. Unqualified samples were discarded.

- **Subsequent reviews:** For every additional 25% completed, the same process as the first review was repeated.

- All data were manually verified, and no LLMs were involved in the annotation or validation process.

### H.3. Annotators

All annotators held a Master's degree and received standardized training organized by the project leads, including task-specific demonstrations and clarification of domain-relevant concepts.

# I. Use of LLMs

In this work, LLMs were used solely to refine and polish the authors' manually written draft for better readability. No LLMs were involved in idea generation.

