# OpenReview forum: "AVI-Bench: Toward Human-like Audio-Visual Intelligence of Omni-MLLMs"
_ICML.cc/2026/Conference — ICML 2026 regular_

### Official Review · Reviewer_agT3 · 2026-03-09

**Soundness:** 4
**Presentation:** 4
**Significance:** 4
**Originality:** 4
**Overall Recommendation:** 4
**Confidence:** 5

**Summary:**

Compared with the previous benchmark, which ignored alignment with human-like audio-visual processing, the paper proposed AVI-Bench to solve this question. To comprehensively evaluate the omni-model, AVI-Bench evaluates the model in three stages: perception, understanding, and reasoning. And the author evaluates 28 omni-models with detailed analysis, providing effective suggestions for model training.

**Compliance With Llm Reviewing Policy:**

Affirmed.

**Key Questions For Authors:**

1. human verification need more description
2. Need more error analysis for SOTA model(Gemini-2.5) for analysis. For example, in reasoning task, which cause Gemini-2.5 fail.
3. data source need more detailed explantation.

**Limitations:**

yes

**Strengths And Weaknesses:**

Strength:
1. Sufficient tasks and metrics to fully evaluate the model.
2. Conducted detailed analysis for omni-model ability
3. four-level classification scheme to for different omni-model intelligence

---

> ### Author Rebuttal · Authors · 2026-03-31
>
> # Response to Reviewer agT3
>
> We thank the reviewer for the positive evaluation and the recognition of AVI-Bench's task and metric completeness, depth of model analysis, and four-level taxonomy.
>
> ---
>
> ### Q1: Human verification needs more description
>
> Thank you for this suggestion. The data quality control pipeline of AVI-Bench is described in Section H (Pipeline and Quality Control). All annotators received standardized training prior to formal annotation, including task definition and annotation guideline briefings, trial annotation on example data with feedback calibration, and inter-annotator consistency checks. In the revised manuscript, we will provide a detailed description of the annotator training procedure in Section H.3 (Annotators).
>
> ---
>
> ### Q2: Error analysis for Gemini-2.5
>
> Thank you for this suggestion. We further examined the failure patterns of Gemini-2.5-Pro on reasoning tasks. The main findings are:
>
> 1. **Directional asymmetry in cross-modal consistency checking**: Our error analysis reveals that in both AVH and VAH tasks, the model may produce false positives (e.g., judging a non-existent entity as present) or exhibit uncertainty (e.g., answering "not sure"). VAH performance is clearly lower than AVH, indicating the model is better at "verifying audio with vision" than "constraining vision with audio," further supporting that audio intelligence remains a bottleneck.
>
> 2. **Fine-grained grounding remains the greatest challenge**: AVLG scores only 21.79. Visualization analysis shows that for cross-modal localization queries (e.g., "the object making the shortest sound duration," "the object making the fastest rhythm"), the predicted bounding boxes may have low overlap with GT boxes, indicating that coarse scene understanding or reasoning is relatively strong but finer cross-modal localization still has a clear gap.
>
> We provide representative failure cases with per-frame visualizations (including GT and predicted bounding box comparisons) via an anonymous link. In the revised manuscript, we will add a Gemini-2.5-Pro error analysis subsection in Section E with corresponding failure case visualizations.
>
> > https://anonymous.4open.science/r/failure_case_figure-6F8F/
>
> ---
>
> ### Q3: Data source needs more detailed explanation
>
> Thank you for pointing this out. AVI-Bench data comes from three sources: (1) fully new construction with manual annotation (62%, 3,614 samples), corresponding to AMIC, VMIC, VAR, AVR, ASQA, VSQA, AVSQA and other tasks; (2) annotation conversion from existing datasets, e.g., AVL/AVLG convert dense masks from AVS-Bench and Ref-AVS into bounding boxes; (3) format restructuring, e.g., AVC, AVH, VAH, AVQA are reorganized from AVHBench, AVCaps, Music-AVQA and other datasets into a unified format. Detailed task-level data sources are listed in Section H. In the revised manuscript, we will add footnotes to Table 1 to clearly define New / Repurposed / Hybrid, with accurate pointers to the appendix for detailed data source descriptions and adaptation steps.

---

> > ### Author Rebuttal · Reviewer_agT3 · 2026-04-03
> >
> > Thank you for the clarifications. I am maintaining my score, as the rebuttal consists primarily of clarifications rather than new evidence.

---

> > > ### Author Response · Authors · 2026-04-03
> > >
> > > We appreciate the reviewer's time and constructive comments. Should you have any further questions, we would be happy to provide additional clarification. We will revise the manuscript in accordance with your suggestions in the revised version. Many thanks!

---

### Official Review · Reviewer_HuQi · 2026-03-10

**Soundness:** 2
**Presentation:** 2
**Significance:** 3
**Originality:** 3
**Overall Recommendation:** 2
**Confidence:** 4

**Summary:**

The authors propose AVI-Bench, a cognitively inspired benchmark designed to evaluate the audio-visual intelligence of Omni-Multimodal Large Language Models (Omni-MLLMs). The benchmark is structured around a progressive evaluation framework spanning four stages based on human cognition: Perception, Understanding, Reasoning, and Primitive Sensation . Furthermore, the authors introduce a four-level hierarchical taxonomy (evaluating Task-, Modality-, Stage-, and Domain-adaptive intelligence) to systematically quantify these human-like capabilities

**Compliance With Llm Reviewing Policy:**

Affirmed.

**Ethical Review Concerns:**

A critical limitation of this work is the lack of deep discussion regarding the data itself. The paper fails to demonstrate whether the dataset coverage for each task is sufficiently comprehensive:

#### **Ambiguity in Primitive Sensation Attributes**: For instance, while the "Primitive Sensation" (PriSe) stage is highlighted as a key contribution , the authors do not explicitly define or discuss which fundamental visual and auditory attributes (e.g., specific frequencies, textures, lighting conditions) are actually covered in these synthetic datasets。

#### **Superficial Dataset Repurposing**: Similarly, for tasks in the other evaluation stages, the methodology gives the impression of merely repurposing existing datasets without deeper justification . A more scientifically rigorous approach would involve first defining the specific real-world scenarios required for each cognitive task, and then explicitly mapping how the chosen datasets fulfill those scenario requirements.

#### **Methodological Imbalance**: Consequently, while the paper's primary strength lies in its innovative task organization and hierarchical evaluation framework , it pays insufficient attention to the intrinsic qualities, diversity, and curation depth of the data powering these evaluations.

**Final Justification:**

After carefully reviewing the authors' responses across two rebuttal rounds, I find myself compelled to lower my initial rating. In my first review, I awarded a high score because I genuinely found the cognitively-inspired perspective meaningful. The introduction of "Primitive Sensation" was presented as a core contribution on par with the AVI-Bench itself, and the paper heavily emphasizes this "human-like" intelligence framework from the title through the main text. However, the subsequent discussions have exposed fatal flaws in this exact foundation.

The core issue is not merely a semantic debate over terminology; it is a fundamental breakdown of the proposed evaluation hierarchy. The categorization of "Primitive Sensation" is highly inaccurate. Strictly speaking, the attributes assigned to this stage (e.g., "beat counting," "temporal order," and "spatial relations") intrinsically require short-term memory, sequence processing, and spatial reasoning. Consequently, this supposed foundational stage already encompasses and conflates the capabilities defined in the subsequent three higher-level stages (Perception, Understanding, and Reasoning).

When I explicitly challenged this structural overlap, the authors sidestepped the issue entirely. Instead of addressing the complex attributes in question, they cherry-picked simpler tasks (e.g., "volume" or "duration") to defend their framework. This evasive response, combined with the persistent conceptual confusion, raises serious concerns about whether the authors possess the requisite interdisciplinary knowledge (bridging cognitive science and computer science) to successfully execute such an ambitious, psychology-grounded framework.

I want to clarify that I do not deny the engineering value of the extensive empirical benchmarking. I had actively considered whether these issues could be resolved through minor revisions. Unfortunately, because the flawed "human-like" narrative is so deeply entangled with the entire manuscript, a quick fix is impossible.


To move forward, the authors are faced with a stark dilemma, and neither option can be resolved through a simple text revision:

- Path 1: Abandon the cognitive science narrative entirely. > If the authors choose this route, they must strip the 'human-like' claims from the title, abstract, and core motivation. They would need to pivot to a purely empirical, computational taxonomy (e.g., categorizing tasks strictly by low-level acoustic/visual feature extraction versus high-level semantic understanding). This approach would unravel the paper's primary conceptual novelty and demand a complete rewrite of the Introduction, motivation, and framework justification.
- Path 2: Rigorously align with interdisciplinary principles. > If the authors insist on keeping their psychology-grounded framework, they must fundamentally redefine the boundaries of Sensation, Perception, Understanding, and Reasoning based on established cognitive science literature. This would force them to redistribute tasks like 'beat counting', 'temporal order', and 'spatial relations' into their correct, higher cognitive tiers. Consequently, the current four-stage evaluation results and rankings for all models would be invalidated, necessitating a complete re-analysis and re-writing of the experimental section under the corrected taxonomy.

In conclusion, both paths demand a fundamental dismantling and reconstruction of the paper's core narrative, methodology, and experimental analysis. Because these foundational flaws cannot be rectified without a massive structural overhaul that goes far beyond the scope of a standard rebuttal or revision phase, I must withdraw my initial support and firmly recommend rejection at this stage.

**Key Questions For Authors:**

#### **AMIC Task Specifics**: Regarding the Audio Multi-instance Classification (AMIC) task in Section 3.2, does the detection of multiple instances merely involve simple counting? Does the dataset explicitly cover challenging scenarios, such as the same individual emitting sounds at different timestamps, or similar but distinct individuals emitting sounds at different times?

#### **Temporal Dependencies**: Section 3.2 notes that the understanding stage evaluates "temporal dependencies". However, none of the examples provided in Figure 1 explicitly depict scenarios that require temporal reasoning. Could you provide examples or clarify exactly how temporal dynamics are evaluated in the benchmark?

#### **Table 1 Clarification**: Could you please explicitly define and explain the meaning of the "Annotation" column (e.g., New, Repurposed, Hybrid) in Table 1?

**Limitations:**

#### **Insufficient Depth in Data Organization and Coverage**: A critical limitation of this work is the lack of deep discussion regarding the data itself. The paper fails to demonstrate whether the dataset coverage for each task is sufficiently comprehensive:

###### **Ambiguity in Primitive Sensation Attributes**: While the "Primitive Sensation" stage is highlighted as a key contribution , the authors do not explicitly define or discuss which fundamental visual and auditory attributes (e.g., specific frequencies, textures, lighting conditions) are actually covered in these synthetic datasets .

###### **Superficial Dataset Repurposing**: For tasks in the other evaluation stages, the methodology gives the impression of merely repurposing existing datasets without deeper justification . A more scientifically rigorous approach would involve first defining the specific real-world scenarios required for each cognitive task, and then explicitly mapping how the chosen datasets fulfill those scenario requirements.

###### **Methodological Imbalance**: Consequently, while the paper's primary strength lies in its innovative task organization and hierarchical evaluation framework , it pays insufficient attention to the intrinsic qualities, diversity, and curation depth of the data powering these evaluations.

**Strengths And Weaknesses:**

## Strengths:

#### **Structured Framework**: The progressive organization of tasks and the construction of the four-level hierarchical evaluation system provide a logical and rigorous methodology for assessing Omni-MLLMs .

#### **Comprehensive Benchmarking**: The paper conducts an extensive and valuable empirical evaluation across 28 existing open-source and closed-source models, offering broad insights into the current state of audio-visual intelligence .

## Weaknesses:

#### **Structural and Narrative Focus (Unclear Hierarchy)**: The Abstract and Introduction present both "AVI-Bench" and the "AVI-Bench-PriSe" extension as dual frameworks right at the beginning, which fragments the core narrative and creates a confusing hierarchy . It is strongly recommended to remove all mentions of the "AVI-Bench-PriSe" nomenclature from the Abstract and Introduction. The entire manuscript should be unified strictly under the single framework of "AVI-Bench". The authors should instead simply highlight that, unlike existing benchmarks, AVI-Bench uniquely introduces an evaluation stage emphasizing "primitive sensation". The current dual-naming approach weakens the coherence of the paper

#### **Misalignment with Cognitive Hierarchy**: The paper claims to be "Humun-like", yet the organizational sequence of the stages contradicts biological processing. If modeled after human intelligence, the sequence should strictly be Primitive Sensation $\rightarrow$ Perception $\rightarrow$ Understanding $\rightarrow$ Reasoning, rather than placing sensation at the end . Furthermore, the cognitive science principles underlying this design are largely relegated to the Appendix . This grounding must be integrated into the main text (especially the Introduction and Discussion) to justify the "human-like" keyword in the title.

#### **Conceptual Conflation in Observation 7**: In Observation 7, equating "primitive sensation" with "unfamiliar domains" is conceptually problematic . The manuscript frequently mixes distinct concepts such as "semantic tasks," "low-semantic," "familiar domains," and "unfamiliar domains." Primitive sensation should be defined by its low-level sensory features (e.g., volume, shape), not strictly by the familiarity of the domain. Additionally, while primitive sensation is discussed, the paper lacks a clear explanation of what distinguishes the primitive sensory inputs across different modalities.

#### **Speculative Claims and Inappropriate Metric Selection (Observation 5)**: The assertion that existing benchmarks "fail to systematically assess the development of AVI across tasks" is overly broad and requires specific justification . Furthermore, in Observation 5, attributing the lower captioning performance of closed-source models to application-specific optimization or a lack of focus is highly speculative and likely flawed due to inappropriate metric selection . The paper relies on traditional n-gram and embedding-based metrics (METEOR, ROUGE_L, CIDEr, SBERT) to evaluate the Audio-Visual Captioning (AVC) task . These traditional metrics are known to unfairly penalize the verbose, descriptive, or conversational output styles typical of modern closed-source LLMs, rendering the conclusion that they perform poorly at captioning highly questionable. To derive referenceable and scientifically sound conclusions, the authors  employ more robust, modern evaluation methods for captioning, such as FENSE or an LLM-as-a-Judge framework

#### **Formatting and Presentation Issues**:

###### Table 1: The "Pub." column is incomplete and inconsistently formatted; it should be properly filled out or removed.

###### References: There are numerous formatting errors in the bibliography . Many entries have missing content or inconsistent journal formats (e.g., the citation for Bubeck et al., "Sparks of artificial general intelligence: Early experiments with gpt-4, 2023" lacks publication venue details

---

> ### Author Rebuttal · Authors · 2026-03-31
>
> # Response to Reviewer HuQi
>
> We thank the reviewer for recognizing AVI-Bench's "logical and rigorous methodology" and the benchmarking of 28 models as "offering broad insights."
>
> ---
> ### W1: Dual naming
> In the revision, we will unify naming under AVI-Bench throughout the abstract and introduction, presenting primitive sensation as the fourth evaluation stage.
>
> ---
> ### W2: Stage ordering vs. biological hierarchy
>
> From a biological cognition perspective, the order should be Sensation → Perception → Understanding → Reasoning. Placing primitive sensation last is a deliberate evaluation design: AVI-Bench is cognition-inspired in *what* to evaluate, but the ordering respects differences between LLMs and human cognition. Low-level perceptual discrimination that is effortless for humans is more challenging for current LLMs, placing it last as a stress test reveals this gap. We chose "human-like" rather than "human-level" to emphasize human cognition as the evaluation reference, not strict biological replication.
>
> We will distinguish biological ordering from evaluation-oriented ordering in Section 3.1, and integrate cognitive science background into Sections 3.
>
> ---
> ### W3: Obs 7 conflates primitive sensation with unfamiliar domain
>
> Primitive sensation evaluates low-level perceptual discrimination (volume, shape, texture). In a familiar domain, models might use semantic priors as shortcuts rather than genuinely discriminating. We chose unfamiliar low-semantic stimuli following the experimental psychology principle of isolating target abilities by removing semantic content (cf. Ebbinghaus [1] using nonsense syllables to isolate memory).
>
> The familiar-vs-unfamiliar comparison then naturally gives rise to L4. The three concepts form a causal chain:
> - Primitive sensation: starting point (what to evaluate)
> - Unfamiliar domain: data choice for fair testing (how to evaluate)
> - Domain-adaptive (L4): naturally derived evaluation dimension
>
> We will clarify this logic in Section 3.2, rewrite Obs 7, and list PriSe attributes:
> - ASQA: beat counting, sound order, volume, clarity
> - VSQA: color, shape, spatial relations, motion, texture
> - AVSQA: sound source identification, temporal order, loudness, duration
>
> >[1] Ebbinghaus, H. (1885). Über das Gedächtnis. Duncker & Humblot.
>
> ---
> ### W4: Observation 5 speculative; AVC metrics
>
> Existing audio-visual benchmarks mostly remain at task-level evaluation without systematic capability grading, we will revise this statement accordingly.
>
> Following the reviewer's suggestion, we conducted FENSE and LLM-as-Judge evaluation for all 28 models. After replacing AVC metrics, taxonomy rankings remain stable: L1 τ=0.989, L2 τ=0.984, L4 τ=0.902, L3 τ=0.730 (p<0.001). The core conclusions hold across metric choices. We will rewrite Observation 5, removing unsupported causal inferences, and add results in Section E.
>
> ---
> ### W5: Table 1 and bibliography
>
> Thanks for pointing this out. Preprint entries will use arXiv to align the format, and AVCaps with new pub info will be updated accordingly. The bibliography will be checked and corrected entry by entry.
>
> ---
> ### Q1: Is AMIC simple counting?
>
> AMIC evaluates both semantic set matching (sound source categories were predicted) and counting accuracy (instance count for each category). The dataset includes scenarios such as different sound sources of the same category emitting sounds at different timestamps. We will supplement a more detailed task description in Section 3.2.
>
> ---
>
> ### Q2: Temporal dependencies in understanding
>
> In the Understanding stage, AVC most directly embodies temporal dependencies: audio-visual events in the video unfold over time, and models need to capture the sequential order of events and cross-modal synchronization to generate accurate descriptions. We will clarify this in the main text.
>
> ---
>
> ### Q3: Annotation column definition
>
> Thank you for pointing this out. We will add footnotes to Table 1 with clear definitions: New (constructed from scratch with manual annotation), Repurposed (reusing existing data reformatted into our unified benchmark format), Hybrid (containing both newly constructed and repurposed data).
>
> ---
>
> ### Limitations
>
> AVI-Bench follows a cognition-first pipeline: we first defined capability requirements per stage (Table 5), then selected or constructed data accordingly. For perception-understanding-reasoning tasks, we selected established datasets (AVS-Bench, Ref-AVS, Music-AVQA, AVHBench) with quality ensured through Section H procedures. For unmet needs (e.g., cross-modal sensation), we built AVSQA from scratch.
>
> We thank the reviewer for this suggestion, which will help AVI-Bench better demonstrate its cognitively inspired design characteristics. In the revision, we will explicitly justify why the dataset was selected and how it satisfies the requirements of the cognitive stage, and detailed table of low-level perceptual attribute coverage for PriSe across modalities (as listed in W3).

---

> > ### Author Rebuttal · Reviewer_HuQi · 2026-04-02
> >
> > I thank the authors for their rebuttal. While the proposed benchmark has significant potential, the response has unfortunately further compounded my concerns regarding its conceptual framework.
> >
> > Even for a computer science paper, maintaining logical consistency and accurate terminology is essential. The current use of "Primitive Sensation" to categorize tasks involving spatial reasoning and memory is scientifically inappropriate. This is not merely a phrasing issue, but a fundamental misalignment that undermines the benchmark’s validity. Furthermore, the persistent inconsistency in core terminology (e.g., volume vs. intensity) and non-standard formatting suggest a lack of rigorous academic standards.
> >
> > In summary, while the core work is promising, the manuscript requires a fundamental reorganization and a major overhaul of the writing to be accurate and readable. Consequently, I find my initial evaluation was too optimistic and will be adjusting my scores to reflect these unresolved issues.

---

> > > ### Author Response · Authors · 2026-04-03
> > >
> > > We sincerely thank Reviewer HuQi for their continued engagement and high standards for terminological precision.
> > >
> > > ---
> > > ### Terminology Clarification
> > > We first acknowledge that in our rebuttal, we inadvertently used "loudness" in the AVSQA attribute list, which was introduced during rebuttal draft polishing for better readability and we apologize for the confusion. The manuscript itself consistently uses "volume" throughout (e.g., main text, figure and appendix). As for the term "intensity" mentioned by the reviewer, we would also like to clarify that this term does not appear in either the paper or the rebuttal; therefore, no inconsistency of this kind is present in our work.
> > >
> > > ---
> > > ### On "spatial reasoning and memory"
> > > We respectfully clarify that PriSe **does not** involve spatial reasoning or memory tasks. Concrete examples from PriSe (illustrated in Figure 2):
> > >
> > > - ASQA: "Which sound has a larger volume?"  [first / last / same ...]
> > > - VSQA: "What is the movement state of the smoothest object?"  [rotation / shrinking / translation ...]
> > > - AVSQA: "Which object is making the sound for the longest duration?"  [cube / pyramid / cone ...]
> > >
> > > These tasks require no reasoning chain or memory retrieval. The spatial content in VSQA/AVSQA is limited to direct detection of spatial position with no inferential steps, e.g., "What is the position of the dot? (top left)". None of these tasks involve reasoning as defined in our framework.
> > >
> > > Regarding citation about "memory": Our response cited Ebbinghaus as a methodological analogy: his use of nonsense syllables to remove semantic content parallels our removal of semantic cues to mitigate models' semantic priors. The citation concerns experimental design inspiration, not the cognitive ability being tested.
> > >
> > > ---
> > > We appreciate the reviewer’s concern for terminological rigor. We will provide clearer annotations for references introduced during the discussion, as well as ensure full terminological consistency in the revised version.
> > >
> > > We also sincerely thank the reviewer for recognizing the significant potential and promise of this work, and for the valuable feedback that will help us further strengthen the manuscript.

---

### Official Review · Reviewer_xcQc · 2026-03-13

**Soundness:** 3
**Presentation:** 3
**Significance:** 4
**Originality:** 3
**Overall Recommendation:** 5
**Confidence:** 4

**Summary:**

This paper introduces AVI-Bench, a benchmark for evaluating the audio-visual intelligence (AVI) of Omni-Multimodal Large Language Models (Omni-MLLMs). The benchmark is organized around three cognitive stages (perception, understanding, and reasoning) and plus an additional "primitive sensation" stage (AVI-Bench-PriSe) that tests models on unfamiliar, low-semantic stimuli. The authors evaluate 28 models, report observations about model capabilities, and propose a four-level taxonomy (task-adaptive, modality-adaptive, stage-adaptive, domain-adaptive) for classifying AVI.

**Compliance With Llm Reviewing Policy:**

Affirmed.

**Final Justification:**

I believe the authors’ responses have addressed my concerns, and I consider this work to have meaningful contributions to the field.

**Key Questions For Authors:**

1. How sensitive are the Level-2 through Level-4 rankings to the choice of α = 0.5?

2. The unimodal ablation (Table 6) reveals that many models perform similarly or better with unimodal inputs on reasoning tasks (AVQA, AVLG). This suggests these tasks may not truly require cross-modal integration for current models. How do you ensure that AVI-Bench is measuring audio-visual synergy rather than strong unimodal reasoning with one dominant modality?

3. Given that not a small portion of the benchmark data is repurposed from existing datasets, what steps were taken to ensure that the evaluated models (especially closed-source ones) have not been trained on overlapping data?

**Limitations:**

yes

**Strengths And Weaknesses:**

Strengths:
- The benchmark covers a broad set of tasks from multi-instance classification and audio-visual localization to hallucination detection and language grounding which goes well beyond what most prior audio-visual benchmarks offer.
- The primitive sensation extension is well-motivated. AVI-Bench-PriSe is a creative and useful addition. Testing whether models can handle low-semantic, unfamiliar stimuli (e.g., detecting volume differences, texture properties) provides insight into whether models have learned general perceptual abilities or merely pattern-matched on training distributions.
- Evaluating 28 models (both open and closed-source) across all tasks strengthens the empirical contribution.

Weaknesses:
- The paper claims to be "cognitively inspired" and maps tasks to brain regions (Table 5), but this mapping is post-hoc and decorative rather than functional.
- Several components of the taxonomy involve ad-hoc design decisions that are insufficiently justified. The scaling constant α = 0.5 in Equation 3 is stated but not motivated.
- Missing analysis of cross-task correlations and redundancy.

---

> ### Author Rebuttal · Authors · 2026-03-31
>
> # Response to Reviewer xcQc
>
> We thank the reviewer for the positive evaluation and the recognition of AVI-Bench's broad task coverage, PriSe extension, and the evaluation of 28 models.
>
> ---
>
> ### W1: Brain region mapping is post-hoc and decorative rather than functional
>
> We need to clarify: AVI-Bench's cognitive stage divisions **are indeed** designed based on human cognition process. We used cognitive levels as the starting point to construct the task system and taxonomy structure, rather than designing tasks first and attaching cognitive labels. As shown in the capability column of Table 5, we first analyzed the function of each level (sensation, perception, understanding, reasoning) and granularity (unimodal vs. cross-modal, local vs. global, coarse vs. finer, etc.).
>
> Human brain regions are complex. The brain region column follows the capability column because our mapping is based on human-like cognition capability rather than physiological structure (brain lobes), as described in Section C.1: "the human brain processes...". We will move the capability and brain lobe columns to the leftmost position, annotate our design rationale in the caption, and in Section C.1 distinguish "cognition function/capability" from "the location where the function/capability occurs/is processed" to prevent ambiguity. The current table places the task ID in the first column for indexing convenience.
>
> ---
>
> ### W2 + Q1: Design motivation and sensitivity of α = 0.5
>
> **Design motivation**: As shown in Eq. (2), the modality discrepancy term Δ_m is defined on [0, 2]. With α = 0.5, the penalty factor (1 - α · Δ_m) covers exactly [0, 1]: no penalty when modalities are perfectly balanced, reducible to 0 under extreme imbalance. Therefore, α = 0.5 is a natural and interpretable normalization choice.
>
> **Sensitivity analysis**: We recomputed the taxonomy rankings for all 28 models under α ∈ {0.25, 0.5, 0.75, 1.0}:
>
> | α | L2 τ | L3 τ | L4 τ | Avg τ | Top-5 Overlap |
> |---|------|------|------|-------|---------------|
> | 0.25 | 0.984 | 0.825 | 0.762 | 0.857 | 4/5 |
> | **0.50** | **1.000** | **1.000** | **1.000** | **1.000** | **5/5** |
> | 0.75 | 0.974 | 0.910 | 0.826 | 0.903 | 4/5 |
> | 1.00 | 0.947 | 0.853 | 0.765 | 0.855 | 4/5 |
>
> Across all α values, avg Kendall's τ > 0.85 and Top-5 overlap is at least 4/5. The relative rankings of the taxonomy are stable across all tested α values.
>
> We will supplement the design motivation for α in Section 5.2 and add the complete sensitivity analysis results in Section E.5.
>
> ---
>
> ### W3: Missing cross-task correlation and redundancy analysis
>
> We conducted a Pearson correlation analysis among the 14 tasks:
>
> - **Within-stage average correlation**: r = 0.591, indicating tasks within the same stage are related but not simply redundant
> - **Cross-stage average correlation**: r = 0.577, supporting the discriminative role of stage design
>
> The two task pairs with the highest correlations are:
> - AVL & AVLG (r = 0.890): sharing spatial grounding capability.
> - AVH & AVQA (r = 0.859): relying on cross-modal understanding capability.
>
> Both high-correlation pairs have reasonable semantic explanations. All other task pairs show clearly lower correlations, indicating that the 14 tasks each contribute different dimensions of diagnostic information.
>
> We will add a 14×14 cross-task correlation heatmap and the above analysis in Section E.6.
>
> ---
>
> ### Q2: How to ensure measuring audio-visual synergy rather than unimodal shortcuts?
>
> AVQA and AVLG involve both audio and visual cues, with their task design targeting cross-modal integration. The fact that some models perform comparably or even better with unimodal inputs precisely indicates that current models' cross-modal fusion capabilities still have significant room for improvement. We thank the reviewer for this insightful observation! We will supplement discussion in Section D.1 to help the community identify on which tasks models already possess cross-modal synergy and on which they still rely on unimodal shortcuts.
>
> ---
>
> ### Q3: Data contamination risk for repurposed data
>
> We have taken multiple layers of mitigation measures:
>
> 1. **62% of data is newly constructed/annotated**, never appearing in any training corpus
> 2. **PriSe data is entirely constructed offline**, without reusing real-world content
> 3. **Repurposed tasks undergo format conversion and question restructuring** (e.g., AVL/AVLG convert dense masks to bounding boxes, AVC/AVH/VAH are unified into new JSON format), reducing the benefit from direct memorization
>
> We also plan to **release evaluation data in encrypted form**, with answers available only through an online evaluation system, reducing the risk of evaluation samples being crawled into training data.
>
> In the revised manuscript, we will add a "Data Contamination Mitigation" subsection in the Appendix to explain these measures.

---

> > ### Author Rebuttal · Reviewer_xcQc · 2026-04-03
> >
> > I believe the authors’ responses have addressed my concerns, and I consider this work to have meaningful contributions to the field.

---

> > > ### Author Response · Authors · 2026-04-03
> > >
> > > Thank you again for your time and thoughtful review. We are very glad to hear that our rebuttal has addressed your concerns, and we greatly appreciate your positive assessment of the paper’s contributions. We will carefully incorporate your suggestions into the final revision.

---

### Official Review · Reviewer_bRtY · 2026-03-13

**Soundness:** 3
**Presentation:** 3
**Significance:** 2
**Originality:** 3
**Overall Recommendation:** 4
**Confidence:** 4

**Summary:**

The authors introduce AVI-Bench, a comprehensive evaluation framework designed to assess the audio-visual intelligence (AVI) of Omni-Multimodal Large Language Models (Omni-MLLMs). The benchmark is structured around a cognitively inspired progression of tasks: perception, understanding, and reasoning. Additionally, it introduces AVI-Bench-PriSe, an extension aimed at testing "Primitive Sensation" using low-semantic and unfamiliar domain stimuli. The study evaluates 28 different models using 13 metrics across 14 diverse tasks

**Compliance With Llm Reviewing Policy:**

Affirmed.

**Key Questions For Authors:**

- In your observation of the modality imbalance, how do you theoretically account for the representation gap between audio and visual modalities? Could you integrate metrics that assess the latent space alignment (e.g., measuring mutual information or representation collapse) alongside your empirical task scores?

- Recent major IQ work such as SpeechIQ (ACL 2025) emphasizes evaluating the intrinsic quality and theoretical bounds of speech representations before they are passed to LLM reasoning engines. How might AVI-Bench incorporate such intermediate representation-level evaluations to explain why reasoning fails in your Level-3 taxonomy? The references depth in the current work seems a bit missing, e.g., also on missing the agentic AudioLM model such as Generative ASR Correction [Neurips 23] and AudioQA.

- Can you provide a more rigorous theoretical discussion on how the specific properties of these encoders (e.g., sequence length, continuous vs. discrete tokenization) influence the cross-modal grounding capabilities measured in the AVL and AVLG tasks?

**Limitations:**

Yes. The authors clearly outline the scope of their benchmark and discuss the limitations of current models , though they could expand on the limitations of their own empirical penalty formulas (e.g., the harmonic mean approach in Level-4 ) versus theoretical alignment bounds.

**Strengths And Weaknesses:**

Pros

- The introduction of AVI-Bench-PriSe to test low-semantic, primitive sensory inputs is a refreshing addition that pushes the boundaries of how we evaluate models beyond their highly curated training distributions.

- Evaluating 28 distinct models (both open and closed-source) across 5,864 samples provides a robust empirical snapshot of the current Omni-MLLM landscape.

Cons

- While the paper successfully identifies an empirical bottleneck in audio intelligence compared to visual intelligence, it lacks a theoretical investigation into why this occurs at the representation level. The transition from continuous audio waveforms to discrete LLM latent spaces introduces quantization errors and alignment challenges that are not theoretically formulated.

- he proposed four-level taxonomy relies on linear combinations and empirical penalties, such as the modality-adaptive score $\mathcal{S}_{M}=(1-\alpha\cdot\Delta_{m})\cdot\mathcal{S}_{T}$. From a representation learning standpoint, it would be highly beneficial to link these empirical scores to information-theoretic bounds (e.g., mutual information between audio and visual representations). As demonstrated in recent literature like SpeechIQ (ACL 2025), quantifying the intrinsic quality and alignment of speech/audio representations provides a much deeper understanding of cross-modal reasoning failure.

- The authors note that replacing weaker audio encoders (e.g., Qwen-Audio with Qwen2-Audio) improves performance. However, the paper does not theoretically analyze how the temporal resolution, frame rate, or latent dimensionality of these encoders impacts the downstream cross-modal grounding tasks (AVL, AVLG) where models severely struggle

---

> ### Author Rebuttal · Authors · 2026-03-31
>
> # Response to Reviewer bRtY
>
> We thank the reviewer for recognizing PriSe as a "refreshing addition" and affirming the "robust empirical snapshot" of 28 models and 5,864 samples.
>
> ---
>
> ### W1 + Q1: Audio-visual representation gap
>
> **(1) Positioning.** AVI-Bench is cognition-inspired in *what* it evaluates (Sensation, Perception, Understanding, Reasoning) and outcome-based in *how* it evaluates (through task outputs). Behavioral performance is itself an externalization of internal representations. Outcome-based evaluation is the mainstream paradigm (MMMU, HLE, ARC, SpeechIQ), and ensures applicability to closed-source models such as Gemini.
>
> **(2) Gap causes.** The audio-visual gap is related to at least: (a) limited pre-training resources for audio encoders compared to visual encoders; (b) less mature community optimization of audio modality integration in Omni-MLLMs. Our multi-stage baseline (Table 8) provides supporting evidence: upgrading only the audio encoder (a1→a2) yields a great gain across all four levels (e.g., L1: 22.52→30.81), confirming that the audio branch is currently the primary bottleneck.
>
> **(3) Next step.** The representation-level analysis proposed by the reviewer (MI, representation collapse) is a natural next step. AVI-Bench provides: (a) diagnostic anchors identifying which modalities, tasks, and stages have shortcomings; (b) a component comparison framework (Table 8) combining different encoders and LLM backbones. For open-source models, researchers can extract modality-specific sub-sequences from multimodal token sequences and probe cross-modal alignment. We will discuss this in Appendix B.4.
>
> ---
>
> ### W2 + Q2: Taxonomy motivation + SpeechIQ
>
> We understand the core concern as: whether AVI-Bench can reveal how earlier-stage quality limits later-stage performance. If inaccurate, we welcome clarification.
>
> **(1) Cross-stage diagnosis.** We studied the mentioned SpeechIQ, which uses an external judge LLM to indirectly assess semantic preservation of ASR outputs. AVI-Bench takes a more direct approach: independent tasks for each cognitive stage, with the four-level taxonomy locating bottlenecks: L1 (task-adaptive) identifies weak tasks, L2 (modality-adaptive) quantifies modality imbalance, L3 (stage-adaptive) reveals cross-stage inconsistencies, and L4 (domain-adaptive) tests performance to unfamiliar domains. For example, Gemini-2.5-Pro: L1=54.54 but L3=20.19, revealing clear stage-level imbalance. Compared to SpeechIQ's 3 levels and speech-only coverage, AVI-Bench provides broader modality coverage and additional diagnostic dimensions.
>
> **(2) On incorporating representation-level evaluation.** The outcome-based design ensures consistent evaluation across all 28 models including closed-source ones, and maintains simplicity in benchmark design. Representation-level analysis is a valuable independent direction and the community can conduct targeted research on open-source models inspired by AVI-Bench's diagnostics.
>
> **(3) References.** We will add SpeechIQ in Section 2.2. Regarding Generative ASR Correction and AudioQA, we could not identify the exact papers and we would be grateful if the DOI or full title could be provided.
>
> ---
>
> ### W3 + Q3: Encoder properties and grounding
>
> The multi-stage baseline (Table 8) provides diagnostic anchors via 8 configurations of audio encoders (a1/a2), visual encoders (v1/v2), and LLM backbones (q1/q2). With other components fixed: audio encoder upgrade (a1→a2) yields L1: 22.52→30.81, L4: 8.58→13.74, clearly show that audio encoder upgrades have a large impact on Omni-MLLM performance.
>
> Since each upgrade couples multiple factors, we cannot attribute the gain to any single attribute (e.g., temporal resolution or tokenization strategy). Precise attribution would require pre-training multiple encoder variants from scratch with all other attributes held constant, computational costs far beyond this work's scope. AVI-Bench provides the evaluation foundation for such future ablation research. We will discuss this in Section D.4.
>
> ---
>
> ### Limitations: Empirical formulas
>
> The taxonomy formulas (e.g., L2's modality-adaptive penalty, L4's harmonic mean) are diagnostic tools designed for practicality and interpretability, providing comparable diagnostic signals for cross-model comparison rather than theoretical alignment bounds. We will clarify their scope in the Limitations section.

---

> > ### Author Rebuttal · Reviewer_bRtY · 2026-04-04
> >
> > thanks for authors' efforts on the rebuttal. I will keep my original recommendation, please add the related agentic based metrics in the final version.

---

> > > ### Author Response · Authors · 2026-04-05
> > >
> > > We thank the reviewer for the helpful comments, and will include the suggested content and experiments in the updated version.

---

### Decision · Program_Chairs · 2026-04-30

**Decision:**

Accept (regular)

**Comment:**

This paper introduces AVI-Bench, a benchmark for evaluating audio-visual intelligence in Omni-MLLMs, with 14 tasks, a four-level taxonomy, and evaluation across 28 models. Reviewers agree the benchmark is broad and empirically valuable, and the PriSe setting for probing low-semantic, unfamiliar stimuli is a strength, though it also raises concerns.

The main issue is the cognitive (“human-like”) framing. The proposed hierarchy (e.g., “Primitive Sensation”) relies on ad-hoc definitions, ambiguous terminology, and limited grounding, and may conflate low-level perception with higher-level processes. This conceptual concern persists and leads to overclaiming relative to what is empirically supported.

Overall, the paper is best viewed as a useful empirical resource (a benchmark and an analysis), largely independent of its cognitive interpretation. The contribution is valuable, but the framing should be significantly softened to improve clarity and avoid overstated claims.